Adding pieces to the puzzle: insights into diversity and distribution patterns of Cumacea (Crustacea: Peracarida) from the deep North Atlantic to the Arctic Ocean

Uhlir Carolin 1 2 carolin.uhlir@senckenberg.de
http://orcid.org/0000-0002-1373-456X Schwentner Martin 1 3
Meland Kenneth 4
Kongsrud Jon Anders 5
Glenner Henrik 4 6
Brandt Angelika 7 8
Thiel Ralf 1
Svavarsson Jörundur 9
http://orcid.org/0000-0001-9942-349X Lörz Anne-Nina 10
http://orcid.org/0000-0002-3269-8904 Brix Saskia 2
1 Center of Natural History (CeNak), Universität Hamburg , Hamburg , Germany
2 German Center for Marine Biodiversity Research (DZMB), Senckenberg Research Institute , Hamburg , Germany
3 Natural History Museum Vienna , Vienna , Austria
4 Department of Biological Sciences, University of Bergen , Bergen , Norway
5 Department of Natural History, University Museum of Bergen (ZMBN) , Bergen , Norway
6 Centre of Macroecology, Evolution and Climate (CMEC), Globe Institute, University of Copenhagen , Copenhagen , Denmark
7 Senckenberg Research Institute and Natural History Museum , Frankfurt am Main , Germany
8 Institute for Ecology, Evolution and Diversity, Goethe University Frankfurt , Frankfurt am Main , Germany
9 Faculty of Life and Environmental Sciences, School of Engineering and Natural Sciences, University of Iceland , Reykjavík , Iceland
10 Institute for Marine Ecosystems and Fisheries Science, Center for Earth System Research and Sustainability (CEN), Universität Hamburg , Hamburg , Germany
Guy-Haim Tamar
Electronic publication date: 2021 Nov 11
Publication date: 2021
Volume: 9
Electronic Location ID: e12379
Received 2021 Apr 29; Accepted 2021 Oct 4
Copyright: © 2021 Uhlir et al.
Copyright year: 2021
Copyright holder: Uhlir et al.
License: This is an open access article distributed under the terms of the Creative Commons Attribution License, which permits unrestricted use, distribution, reproduction and adaptation in any medium and for any purpose provided that it is properly attributed. For attribution, the original author(s), title, publication source (PeerJ) and either DOI or URL of the article must be cited.
License URL: https://creativecommons.org/licenses/by/4.0/

Keywords: Species delimitation, Integrative taxonomy, IceAGE project, 16S rDNA gene, Iceland, Benthic fauna, Deep sea, Biogeography

Funding: Universität Hamburg Universität Hamburg and the Center of Natural History Norwegian Taxonomic Initiative (NTI) Angelika Brandt and Saskia Brix AWI_PS106_00 DFG BR3843/3-1 Carolin Uhlir received a grant from the Universität Hamburg to travel to Bergen. Lab work was funded via internal funds of the Universität Hamburg and the Center of Natural History. Kenneth Meland and Henrik Glenner were supported by the Norwegian Taxonomic Initiative (NTI). Angelika Brandt and Saskia Brix received grant support from AWI_PS106_00 for the PASCAL and from the DFG BR3843/3-1 for the IceAGE expedition. The funders had no role in study design, data collection and analysis, decision to publish, or preparation of the manuscript.

==============================
The Nordic Seas have one of the highest water-mass diversities in the world, yet large knowledge gaps exist in biodiversity structure and biogeographical distribution patterns of the deep macrobenthic fauna. This study focuses on the marine bottom-dwelling peracarid crustacean taxon Cumacea from northern waters, using a combined approach of morphological and molecular techniques to present one of the first insights into genetic variability of this taxon. In total, 947 specimens were assigned to 77 morphologically differing species, representing all seven known families from the North Atlantic. A total of 131 specimens were studied genetically (16S rRNA) and divided into 53 putative species by species delimitation methods (GMYC and ABGD). In most cases, morphological and molecular-genetic delimitation was fully congruent, highlighting the overall success and high quality of both approaches. Differences were due to eight instances resulting in either ecologically driven morphological diversification of species or morphologically cryptic species, uncovering hidden diversity. An interspecific genetic distance of at least 8% was observed with a clear barcoding gap for molecular delimitation of cumacean species. Combining these findings with data from public databases and specimens collected during different international expeditions revealed a change in the composition of taxa from a Northern Atlantic-boreal to an Arctic community. The Greenland-Iceland-Scotland-Ridge (GIS-Ridge) acts as a geographical barrier and/or predominate water masses correspond well with cumacean taxa dominance. A closer investigation on species level revealed occurrences across multiple ecoregions or patchy distributions within defined ecoregions.

Introduction

The ocean surrounding Iceland and its adjacent waters have one of the world’s highest diversities of water masses (Hansen & Østerhus, 2000). The hydrography of the area is rather complex as several primary water masses meet and often overlay each other (Malmberg & Valdimarsson, 2003; Brix & Svavarsson, 2010; Meißner, Brenke & Svavarsson, 2014). According to these hydrographic features, benthic habitats are characterized by depth gradients, water-mass parameters and habitat structure (Meißner, Brenke & Svavarsson, 2014). Thus, environmental data is important to help understand the driving forces of species’ distribution patterns.

It is widely accepted that ‘Arctic’ water masses are distinguished from ‘Subarctic’ water masses by their origin from the upper 250–300 m of the Arctic Ocean, whereas the latter describe a mixture of polar and non-polar (Atlantic or Pacific) water masses (Dunbar, 1951, 1972; Curtis, 1975). Composition and distribution of benthic organisms in the Arctic Ocean is related to water masses, but also to the geological history (Bluhm et al., 2011; Mironov, Dilman & Krylova, 2013). The Fram Strait between North-East Greenland and Svalbard is the only deep-reaching connection to the Arctic Basin (sill depth > 2,200 m). In the Icelandic region, the Greenland-Iceland-Scotland Ridge (GIS-Ridge) is a natural border for benthic organisms extending from East Greenland to Scotland and forming a continuous barrier between the North Atlantic, the North European and Siberian Seas and the Arctic Ocean north of the ridge (Hansen & Østerhus, 2000). It acts as a transition region exhibiting major temperature differences between water masses of the warmer North Atlantic and colder Greenland, Iceland and Norwegian Sea (GIN-Seas, also termed the Nordic Seas; Brix et al., 2018a). Gaps along this ridge allow deep-water exchange between East Greenland and Iceland across the Denmark Strait and the Faroe Bank Channel between the Faroe Islands and the Faroe Bank, which, at 860 m, is the deepest connection between the >4,000 m deep basins separated by the GIS-Ridge (Brix & Svavarsson, 2010). Earlier studies in this region revealed a trend of north-south separation of benthic crustacean species distributions (Weisshappel & Svavarsson, 1998; Weisshappel, 2000; Weisshappel, 2001) and further outlined the ridge as a potential pathway for the dispersal of shelf fauna from Norway towards Iceland (Brix et al., 2018a).

Crustaceans of the taxon Peracarida Calman, 1904 often form a major fraction of macrobenthic communities in terms of diversity and abundance in Arctic and Subarctic waters (Brandt, 1997; Conlan et al., 2008; Stransky & Svavarsson, 2010). They are characterized by a marsupium, a brood pouch on the ventral side of the carapace of the mature female (Westheide & Rieger, 1996; Silva, 2016). Juveniles hatch as a manca stage by skipping the planktonic stage. In this study, we will focus on the peracarid taxon Cumacea Krøyer, 1846, which are primarily marine bottom-dwelling benthic crustaceans, spending most of their life buried in or close to the sediment with an adapted morphology for a sediment-water-interface lifestyle. Thus, cumaceans are assumed to be restricted in their dispersal abilities and are most likely not able to drift over vast distances (Rex, 1981; Wilson & Hessler, 1987).

Most species have a specialized feeding strategy as detritus or filter feeders. Some more derived taxa have evolved in association with other epibenthic organisms such as sponges or corals and established a strategy as that of scavengers and micro-predators (e.g., Campylaspis G. O. Sars, 1865) with modified mouth parts as piercing organs (Foxon, 1936; Jones, 1976; Petrescu et al., 2009).

Currently there are over 1,800 accepted cumacean species recorded worldwide categorized into eight families (Watling & Gerken, 2019). Approximately 250 cumacean species are recorded in the high-latitude Arctic regions and at least 19 species are known as Arctic endemic species (Vassilenko, 1989). According to the most recent studies on biogeographical patterns of cumaceans in respect to water masses in the Arctic, the families Diastylidae Bate, 1856 and Nannastacidae Bates, 1966 are the most species rich and most widely distributed (Vassilenko, 1989; Watling & Gerken, 2005). The family Leuconidae G. O. Sars, 1878 is the second most species rich taxon and commonly found in colder waters (Vassilenko, 1989; Haye, Kornfield & Watling, 2004; Watling & Gerken, 2005). The predominantly warm-water family Bodotriidae Scott, 1901 and temperate cold-water family Lampropidae G. O. Sars, 1878 contain fewer representatives, but also some endemic Arctic species. Vassilenko (1989) divided the cumacean fauna in the Arctic Ocean into six biogeographic groups, listed in order of decreasing number of species: Boreal-Arctic, Arctic, Atlantic boreal, Pacific boreal, Atlantic subtropical-boreal and Amphiboreal species. In a later publication (Vassilenko, 2002), the Arcto-Atlantic bathyal species group was added to include widespread species from North Atlantic intermediate to near-bottom Arctic water at the continental slope of Arctic Ocean. A complete species list of biogeographic species’ distributions is provided by Vassilenko (1989), Vassilenko & Brandt (1996), Watling & Gerken (2005) and Watling (2009). A reference catalogue of previous studies of the cumacean fauna in North Atlantic and the Atlantic sector of the Arctic Ocean is presented in Vassilenko (1989).

In Subarctic and Arctic Ocean regions, the typically patchy distribution patterns of many cumacean species correspond well with the distribution of major water masses (Gerken & Watling, 1999; Gage et al., 2004; Watling, 2009), as well as local sediment grain size as most cumaceans feed by scraping sand grains (Foxon, 1936). Distribution patterns are less controlled by depth; thus, most species are not restricted to deep-sea areas (Hansen, 1920; Haye, 2002; Watling & Gerken, 2005). The same pattern is assumed for another peracarid taxon, Tanaidacea Dana, 1849 (Błażewicz-Paszkowycz & Siciński, 2014), whereas species distributions of Isopoda Latreille, 1817 seem to be mostly driven by depth and related factors (Schnurr et al., 2014; Brix et al., 2018b). A recent study by Lörz et al. (2021) about amphipods supports water-mass properties to be the main factor shaping species distributions at the boundary between the North Atlantic and Arctic waters as well as the prominent submarine Greenland-Iceland-Faroe Ridge playing a major role in hindering the exchange of deep-sea species between northern and southern deep-sea basins. Large numbers of cumaceans are assumed to remain undiscovered in greater depths, as shelf fauna has been studied to a larger extent and, thus, the abundance and diversity of cumaceans is probably underestimated (Jones & Sanders, 1972; Vassilenko, 1989; Gage et al., 2004 and references therein).

This study aims to present a first insight into biogeographical species diversity of cumaceans from North Atlantic to Arctic waters. The integration of species occurrence records from public databases such as the Global Ocean Biogeographic Information System (OBIS) and the Marine Area database for Norwegian waters (MAREANO) will build the baseline for a species catalogue in the investigated area. New occurrence records provided by the present study will contribute to a better understanding of species distribution ranges for future research on cumacean distribution patterns. Morphological and molecular techniques are used for an integrative taxonomy approach and will increase the knowledge of genetic and morphological variability of this understudied taxon.

Materials & methods

Sampling and study-area properties

The study area includes the northernmost part of the North Atlantic, extending across the GIN-Seas up to the Arctic Ocean. The main bulk of specimens included in this study was collected during the following international projects and expeditions: IceAGE (Icelandic marine Animals: Genetics and Ecology; Cruise M85/3 in 2011; Brix et al., 2014a; Meißner et al., 2018), which is a follow up of the BIOFAR (Biology of the Faroe Islands; Nørrevang et al., 1994; Gerken & Watling, 1999) and the BIOICE project (Benthic Invertebrates of Icelandic waters; Omarsdottir et al., 2013), and PASCAL (Physical feedbacks of Arctic PBL, Sea ice, Cloud and Aerosol; Cruise PS106/1 in 2017; Macke & Flores, 2018) onboard the RVs Meteor and Polarstern, focusing on remote shelf-break and deep-sea habitats within a depth range of 579–2,748 m (Fig. 1). Grant support and field permits are available under BR3843/3-1 and AWI_PS106_00. Additional specimens from the Norwegian Sea and waters off Svalbard sampled by the MAREANO program (Thorsnes, 2009) and the University of Bergen were included (Table 1). Cumaceans were sampled in large amounts in all projects.

Figure 1 Station sites of investigated cumacean specimens sampled during IceAGE and PASCAL expedition.

(A) All investigated station sites of cruise leg M85/3 (IceAGE) and PS106/1 (PASCAL) with information on the study area, deployed gear types and assigned water masses after Schlichtholz & Houssais (2002), Hansen & Østerhus (2000), Brix & Svavarsson (2010) and Ostmann, Schnurr & Martínez Arbizu (2014). (B) Drifting area of cruise leg PS106/1 marking the seven Box corer stations (BC; yellow stars) and the one Epibenthic sled station (EBS; green star; Macke & Flores, 2018).

Table 1 Information on projects and sampled stations within the study area of Greenland, Iceland and Norwegian Sea (GIN-Seas).

Information (if applicable) include start and end position of gear deployment (Latitude/Longitude) and calculated haul distance, deployed gear type, depth, CTD-data on near bottom temperature and salinity, drift velocity of the vessel during PASCAL and assigned water masses after Schlichtholz & Houssais (2002), Hansen & Østerhus (2000), Brix & Svavarsson (2010) and Ostmann, Schnurr & Martínez Arbizu (2014).

Project/Expedition	Station	Area	Region	Habitat	Date	Latitude start	Longitude start	Latitude end	Longitude end	Gear	Depth (m)	Temp. bottom (°C)	Salinity bottom (ppt)	Haul distance (m)	Drift velocity (kn)	Water mass	
Alaska	90,626	N/A	Juneau (Alaska)	shelf	26.06.2009	58.37,690	−134.56,69	N/A	N/A	RP-EBS	1	N/A	N/A	N/A	N/A	Alaska Coast	
BIOICE	BIOICE3669	N/A	Iceland	shelf	25.04.2004	66.19,930	−23.30,780	N/A	N/A	RP-EBS	158	N/A	N/A	N/A	N/A	Modified North Atlantic Water (MNAW)	
IceAGE	961	1	South Iceland Basin	deep sea	28.08.2011	60.0455	−21.50233	N/A	N/A	BC	2,748	2.53	34.99	N/A	N/A	ISOW	
IceAGE	983	3	South Iceland Basin	deep sea	30.08.2011	60.35733	−18.135666	60.0455	−18.14183	RP-EBS	2,568	2.66	35	2,462	N/A	ISOW	
IceAGE	1010	6	South Iceland Basin	slope	02.09.2011	62.55166	−20.39516	62.55366	−20.38116	RP-EBS	1,385	3.88	35.02	N/A	N/A	NAW	
IceAGE	1057	11	South Iceland Irminger Basin	deep sea	07.09.2011	61.64166	−31.35616	61.654	−31.34916	RP-EBS	2,505	3.16	34.94	1,983	N/A	LSW	
IceAGE	1072	13	South Iceland Irminger Basin	deep sea	08.09.2011	63.00766	−28.06816	63.01833	−28.0525	RP-EBS	1,594	4.28	34.99	1,673	N/A	NAW	
IceAGE	1123	19	East Greenland Denmark Strait	slope	14.09.2011	67.21383	−26.2075	67.21466	−26.19216	RP-EBS	716.5	0.07	34.91	670	N/A	APW/NSAIW	
IceAGE	1136	20	East Greenland Denmark Strait	shelf	14.09.2011	67.63583	−26.7665	67.63266	−26.77366	CliSAP Sled	316	0.71	34.63	366	N/A	APW	
IceAGE	1144	21	East Greenland Denmark Strait	deep sea	15.09.2011	67.86783	−23.69633	67.8595	−23.69616	RP-EBS	1,281	−0.67	34.91	1,340	N/A	NSDWw	
IceAGE	1153	22	North-East Iceland Norwegian Basin	deep sea	17.09.2011	69.09333	−9.9335	N/A	N/A	BC	2,174	−0.75	34.91	N/A	N/A	NSDWw	
IceAGE	1155	22	North-East Iceland Norwegian Basin	deep sea	17.09.2011	69.11483	−9.912	69.10616	−9.9205	Brenke Sled	2,204	−0.75	34.91	1,582	N/A	NSDWw	
IceAGE	1184	26	Norwegian Sea Basin	deep sea	20.09.2011	67.64383	−12.162	67.63866	−12.138	RP-EBS	1,819	−0.85	34.91	1,885	N/A	NSDWc	
IceAGE	1191	27	North-East Iceland Norwegian Sea	deep sea	21.09.2011	67.07866	−13.06383	67.07516	−13.03816	RP-EBS	1,575	−0.74	34.91	1,795	N/A	NSDWw	
IceAGE	1219	30	East Iceland Norwegian Sea	slope	22.09.2011	66.289	−12.347	66.2925	−12.355	RP-EBS	579	−0.4	34.9	760	N/A	NSAIW	
MAREANO	R488-379, BT	N/A	Eggakanten (Norway)	deep sea	10.10.2009	69.43,760	15.11,110	N/A	N/A	Beam Trawl	2,241	N/A	N/A	N/A	N/A	Norwegian Coastal Current	
MAREANO	R721-126, RP	N/A	Continental shelf (Norway)	shelf	25.07.2011	67.50,810	11.48,850	N/A	N/A	RP-EBS	183	N/A	N/A	N/A	N/A	Norwegian Coastal Current	
MAREANO	R754-132, RP	N/A	Continental slope (Norway)	slope	22.09.2011	67.48,275	09.41,126	N/A	N/A	RP-EBS	823	N/A	N/A	N/A	N/A	Warm Atlantic Current	
MAREANO	R814-22, RP	N/A	Continental shelf (Norway)	shelf	11.05.2012	67.39,220	10.18,580	N/A	N/A	RP-EBS	224	N/A	N/A	N/A	N/A	Norwegian Coastal Current	
PASCAL	22-3	N/A	Yermak Plateau	deep sea	05.06.2017	81.93248	10.959499	N/A	N/A	BC	1,077	−0.55	34.91	N/A	0.1	NSDWw	
PASCAL	24-5	N/A	Yermak Plateau	slope	07.06.2017	81.927034	10.13311	81.921801	10.055294	EBS	955	−0.48	34.92	1,345	0.3	NSDWw	
PASCAL	25-5	N/A	Yermak Plateau	slope	08.06.2017	81.896594	9.855325	N/A	N/A	BC	931	N/A	N/A	N/A	0	NSDWw	
PASCAL	29-4/5	N/A	Yermak Plateau	deep sea	12.06.2017	81.820493	11.566229	N/A	N/A	BC	1,564	−0.74	34.92	N/A	0.1	NSDWw	
PASCAL	29-7	N/A	Yermak Plateau	deep sea	12.06.2017	81.815543	11.54354	N/A	N/A	BC	1,569	−0.74	34.92	N/A	0.1	NSDWw	
PASCAL	30-1	N/A	Yermak Plateau	deep sea	13.06.2017	81.822024	11.538371	N/A	N/A	BC	1,547	−0.72	34.92	N/A	0.1	NSDWw	
PASCAL	32-3	N/A	Yermak Plateau	deep sea	15.06.2017	81.728044	10.851854	N/A	N/A	BC	1,581	−0.7	34.92	N/A	0.1	NSDWw	
PASCAL	32-4	N/A	Yermak Plateau	deep sea	15.06.2017	81.720718	10.811252	N/A	N/A	BC	1,543	−0.7	34.92	N/A	0.2	NSDWw	
UoB	09.01.28-2	N/A	Fanafjorden (Norway)	shelf	28.01.2009	60.16,440	05.11,050	N/A	N/A	RP-EBS	180	N/A	N/A	N/A	N/A	Fjord	
UoB	11.01.19-1	N/A	Fensfjorden (Norway)	shelf	19.01.2011	60.50,856	04.51,702	N/A	N/A	RP-EBS	460	N/A	N/A	N/A	N/A	Fjord	
UoB	11.01.21-1	N/A	Hjeltefjorden (Norway)	shelf	21.01.2011	60.37,600	04.52,400	N/A	N/A	RP-EBS	205	N/A	N/A	N/A	N/A	Fjord	
UoB	11.03.09-1	N/A	Fensfjorden (Norway)	shelf	09.03.2011	60.51,935	04.54,384	N/A	N/A	RP-EBS	462	N/A	N/A	N/A	N/A	Fjord	
UoB	11.03.11-2	N/A	Hjeltefjorden (Norway)	shelf	11.03.2011	60.37,320	04.52,794	N/A	N/A	RP-EBS	239	N/A	N/A	N/A	N/A	Fjord	
UoB	11.05.10-1	N/A	Sognesjøen (Norway)	shelf	10.05.2011	60.55,048	04.38,225	N/A	N/A	RP-EBS	550	N/A	N/A	N/A	N/A	Norwegian Coastal Current	
UoB	11.05.10-3	N/A	Sognesjøen (Norway)	shelf	10.05.2011	60.57.986	04.40.912	N/A	N/A	RP-EBS	381	N/A	N/A	N/A	N/A	Norwegian Coastal Current	
UoB	11.05.11-2C	N/A	Førdefjorden (Norway)	shelf	11.05.2011	61.29,106	05.21,497	N/A	N/A	RP-EBS	335	N/A	N/A	N/A	N/A	Fjord	
UoB	11.05.15-1	N/A	Hjeltefjorden (Norway)	shelf	15.05.2011	60.37,600	04.52,300	N/A	N/A	RP-EBS	240	N/A	N/A	N/A	N/A	Fjord	
UoB	BS 14-19	N/A	Skagerak (Norway)	shelf	13.05.2009	58.22,101	10.20,572	N/A	N/A	RP-EBS	407	N/A	N/A	N/A	N/A	Mixed Atlantic/Baltic	
UoB	BS 22-32	N/A	Skagerak (Norway)	shelf	14.05.2009	58.28,908	10.26,612	N/A	N/A	RP-EBS	301	N/A	N/A	N/A	N/A	Mixed Atlantic/Baltic	
UoB	BS 28-44	N/A	Skagerak (Norway)	shelf	14.05.2009	58.37,710	10.22,558	N/A	N/A	RP-EBS	251	N/A	N/A	N/A	N/A	Mixed Atlantic/Baltic	
UoB	BS 34-56	N/A	Skagerak (Norway)	shelf	15.05.2009	58.28,172	10.07,999	N/A	N/A	RP-EBS	513	N/A	N/A	N/A	N/A	Mixed Atlantic/Baltic	
UoB	BS 75-135	N/A	Skagerak (Norway)	shelf	19.05.2009	58.51,456	10.26,348	N/A	N/A	RP-EBS	246	N/A	N/A	N/A	N/A	Mixed Atlantic/Baltic	
UoB	BS 82-147	N/A	Skagerak (Norway)	shelf	20.05.2009	58.37,258	10.03,077	N/A	N/A	RP-EBS	484	N/A	N/A	N/A	N/A	Mixed Atlantic/Baltic	
UoB	BS 86-151	N/A	Skagerak (Norway)	shelf	20.05.2009	58.25,540	09.38,836	N/A	N/A	RP-EBS	710	N/A	N/A	N/A	N/A	Mixed Atlantic/Baltic	
UoB	H2DEEP-RP-1	N/A	Jan Mayen	deep sea	05.08.2009	75.35,340	07.45,390	N/A	N/A	RP-EBS	2,542	N/A	N/A	N/A	N/A	N/A	
UoB	KV-09 2011	Sektor 4	Continental shelf (Norway)	shelf	03.06.2011	61.08376	2.49373	N/A	N/A	Grab	191	N/A	N/A	N/A	N/A	Norwegian Coastal Current	
UoB	SFND-08R 2011	N/A	Continental shelf (Norway)	shelf	31.05.2011	61.48140	1.85231	N/A	N/A	Grab	273	N/A	N/A	N/A	N/A	Norwegian Coastal Current	
UoB	UNIS 2007-129	N/A	Svalbard	shelf	04.09.2007	80.05,008	22.11,834	N/A	N/A	RP-EBS	188	N/A	N/A	N/A	N/A	NSDWw	
UoB	UNIS 2009-27	N/A	Svalbard	shelf	01.09.2009	80.09,141	16.56,126	N/A	N/A	RP-EBS	340	N/A	N/A	N/A	N/A	NSDWw	
UoB	UNIS 2009-36	N/A	Svalbard	shelf	01.09.2009	79.36,693	18.55,051	N/A	N/A	RP-EBS	337	N/A	N/A	N/A	N/A	NSDWw	
UoB	UNIS 2009-4	N/A	Svalbard	shelf	25.08.2009	78.18,300	14.29,000	N/A	N/A	Dredge	56	N/A	N/A	N/A	N/A	NSDWw	
UoB	UNIS 2009-71	N/A	Svalbard	shelf	04.09.2009	80.27,453	12.20,208	N/A	N/A	RP-EBS	497	N/A	N/A	N/A	N/A	NSDWw	
UoB	UNIS 2009-73	N/A	Svalbard	shelf	04.09.2009	80.27,410	12.46,836	N/A	N/A	RP-EBS	452	N/A	N/A	N/A	N/A	NSDWw	
UoB	UNIS 2007-140	N/A	Svalbard	shelf	04.09.2007	80.38,915	22.06,067	N/A	N/A	RP-EBS	126	N/A	N/A	N/A	N/A	NSDWw	
UoB	VGPT1-22 2009	Sektor 4	Continental shelf (Norway)	shelf	29.05.2009	61.23,724	02.07,254	N/A	N/A	Grab	295	N/A	N/A	N/A	N/A	Norwegian Coastal Current	
UoB	VGPT2-10 2011	N/A	Continental shelf (Norway)	shelf	28.05.2011	61.22,632	02.06,582	N/A	N/A	Grab	284	N/A	N/A	N/A	N/A	Norwegian Coastal Current	
UoB	VI-22 2011	Sektor 4	Continental shelf (Norway)	shelf	26.05.2011	61.22,140	02.26,688	N/A	N/A	Grab	334	N/A	N/A	N/A	N/A	Norwegian Coastal Current	

Temperature and salinity were considered as hydrographic variables for the evaluation of water-mass characteristics, which were available for IceAGE areas (http://www.vliz.be/en/imis?module=dataset&dasid=6211) and PASCAL stations (https://doi.pangaea.de/10.1594/PANGAEA.881579) measured just off the sea floor with a conductivity-temperature-depth profiler (CTD) (see Brix et al. (2012); Brix & Devey (2019)). Each area and station was allocated to a defined water mass according to the definition of Schlichtholz & Houssais (2002), which is applicable for the Fram Strait region and, thus, the entrance to the deep Arctic Eurasian Basin. For the GIN-Seas around Iceland, the definitions as described by Hansen & Østerhus (2000), Brix & Svavarsson (2010) and Ostmann, Schnurr & Martínez Arbizu (2014) have been used as baseline (Table S1). The records were manually divided into eight ecoregions based on their predominate water-mass characteristics after the combined definitions of Curtis (1975), Spalding et al. (2007) and Piepenburg et al. (2011): Warm North Atlantic water mass (North Atlantic Ocean, ecoregion 4), intermediate Subarctic water mass (East Greenland Sea, 2; Norwegian Sea, 5; Barents Sea, 7) and cold Arctic water mass (Arctic Basin, 1; Kara Sea, 8; North Greenland Sea, 3; White Sea, 6).

Sampling data and sample treatment

Specimens were obtained using different types of benthic sampling gear. Most frequently applied was the Rothlisberg-Pearcy Epibenthic sled (RP-EBS, Rothlisberg & Pearcy, 1976; Brattegard & Fosså, 1991), equipped with a net of 500 µm mesh size and ending in a collecting cod end of 300 µm mesh size. Different from the standard deployment protocols as outlined in Brenke (2005), the sampling during the PASCAL expedition was conducted while the vessel was attached to an ice floe during a 2-week passive drifting according to the ocean’s current with an average drift velocity of 0.12 kn. Furthermore, the Camera-Epibenthic sled (C-EBS; Brandt et al., 2013), the Brenke-Sled (Brenke, 2005) and the Giant Box corer (BC; Hessler & Jumars, 1974) were deployed. A detailed description of the sampling design is given in Brix et al. (2014a). Once the deployed gear was on board, the haul was carefully floated in seawater and evenly decanted gently over a series of sieves with mesh sizes of 1/0.5/0.3 mm, washed with sea water on a sieving table and bulk-fixed in precooled 96% undenatured ethanol. All samples were treated as described in Riehl et al. (2014), ensuring that the samples stayed consistently cooled. The samples were sorted either directly on board or afterwards in the laboratories of the German Center for Marine Biodiversity Research (DZMB, Senckenberg am Meer, Hamburg, Germany).

Morphological specimen identification

A total of 947 specimens (Table S2) were determined to the lowest possible taxonomic rank, based primarily on original species descriptions (e.g., Hansen, 1920; Sars, 1900). Species identifications were conducted at the Department of Biological Sciences (University of Bergen, Norway) and DZMB Hamburg using either a ZEISS SteREO Discovery V8 or Leica MZ12.5 dissecting microscope. Dissected pereopods and mouth parts were assessed under a ZEISS Primo Star compound microscope. High quality pictures with depth of focus were taken with a Leica DFC400 digital compound microscope camera using the Z-stacking option in the Leica Application Suite imaging software. Current authoritative classification follows the catalogue World Cumacea Database (http://www.marinespecies.org/cumacea/, Watling & Gerken, 2019) in the World Register of Marine Species (WoRMS Editorial Board, 2019). Additionally, comparative museums’ material has been obtained from the Center of Natural History Hamburg (CeNak) and the University Museum of Bergen (ZMBN).

Molecular methods

DNA extraction, PCR amplification and sequencing

To delimit putative species genetically, DNA extraction and PCR amplification were performed in the laboratories of UoB in Bergen and CeNak in Hamburg. To ascertain that a morphological voucher retained intact, DNA extraction was only performed if at least two individuals were morphologically assigned to the same species. Three different manual workflow kits (DNeasy® Blood and Tissue Kit, QIAGEN®; E.Z.N.A.® Mollusc DNA Kit, Omega Bio-tek, Inc., Norcross, GA, USA; Marine Animal Tissue Genomic DNA Extraction Kit, Neo-Biotech, Pasadena, CA, USA) and one chelating resin (Chelex® 100; Bio-Rad Laboratories, Hercules, CA, USA) were used by following the manufacturer’s instructions, except for the subsequent cleanup step within the DNeasy Blood and Tissue Kit, which was conducted using the AMPure XP beads, ©Beckman Coulter.

All DNA extracts were stored immediately after processing at −20 °C. Nucleic acid concentration (ng/µl) and purity of one µL DNA extract was measured with a Thermo Scientific NanoDrop™ 2,000 Spectrophotometer for all extractions. When the measured concentration exceeded 20 ng/µl, DNA template was diluted 1:10 with ddH2O.

PCR reactions were performed in a reaction volume of 15 µL, consisting of 0.05 µL DreamTaq DNA Polymerase, 1.5 μL DreamTaq Buffer (Thermo Fisher Scientific, Germany), 0.12 μl dNTPs mix (25 mM each), 1.5 μL of each primer (10 mM each) and 1–2 µL DNA extract. Two different sets of 16S rRNA gene primers were utilized, 16Sar-L (5′-CGCCTGTTTATCAAAAACAT-3′, Palumbi, Martin & Romano, 1991) and 16Sb (5′-CTCCGGTTTGAACTCAGATCA-3′, Xiong & Kocher, 1991), which was particularly successful for species of the families Diastylidae, Lampropidae and Leuconidae, and 16SALh (5′-GTACTAAGGTAGCATA-3′) and 16SCLr (5′-ACGCTGTTAYCCCTAAAGTAATT-3′, Rehm, 2007; Rehm et al., 2020), which yielded better results for the Bodotriidae, the Ceratocumatidae Calman, 1905 and some Nannastacidae. However, the latter results in a ~ 200 bp shorter DNA fragment, thus these short sequences were included only in the phylogenetic analyses and were excluded from genetic distance analyses. PCR program had a reaction profile of 94 °C (2 min.), 38 cycles of 94 °C (20 s), 46 °C (10 s) and 65 °C (1 min.) and final extension step of 65 °C (8 min.) was applied. PCR products were purified by incubating 11–13 µL PCR product with 0.8 µL FastAP (one U/µL) and 0.4 µL Exonuclease I (20 U/µL) (both Thermo Fisher Scientific™, Waltham, MA, USA) in 37 °C for 15 min and 80 °C for 15 min. Bidirectional sequencing was performed with the respective PCR primer set, either with Macrogen Europe, Inc (Amsterdam-Zuidoost, Netherlands) or Eurofins Genomics Germany GmbH. Out of 123 extracted specimens, 80 yielded sequence data of sufficient quality to be included in the molecular species delimitation (Table 2). These sequences can be accessed via GenBank and BoLD (dx.doi.org/10.5883/DS-ICECU).

Table 2 Information on cumacean specimens included in the molecular species delimitation based on 16S rRNA gene region sequences.

Species ID groups specimens assigned morphologically to the same species and letters (A–C) show separated lineages by genetic analyses. Sequence ID identifies each specimen in the conducted phylogenetic analyses. Outgroup sequences of other peracarids (Amphipoda, Isopoda, Tanaidacea) are highlighted in grey.

Species ID	Project	Station	Sample ID	GenBank Accession	Higher taxon	Putative species	Sequence ID (Field ID)	
NA	NA	NA	NA	HQ450558	Bodotriidae	Atlantocuma sp.	HQ450558	
Bod01	IceAGE	983	DZMB-HH-68412	MZ402659	Bodotriidae	Bathycuma brevirostre	ICE1-Bod004	
Bod03	IceAGE	1072	DZMB-HH-68361	MZ402660	Bodotriidae	Bodotriidae sp. 1	ICE1-Bod003	
NA	NA	NA	NA	AJ388111	Bodotriidae	Cumopsis fagei	AJ388111	
Bod05-B	IceAGE	983	DZMB-HH-68410	MZ402681	Bodotriidae	Cyclaspis longicaudata	ICE1-Bod001	
Bod05-B	IceAGE	983	DZMB-HH-68411	MZ402680	Bodotriidae	Cyclaspis longicaudata	ICE1-Bod002	
Bod05-A	UoB	11.05.15-1	Bio material=4-6	MK613872.1	Bodotriidae	Cyclaspis longicaudata	seq2	
Bod05-A	UoB	VI-22 2011	Bio material=7-8	MK613873.1	Bodotriidae	Cyclaspis longicaudata	seq3	
NA	NA	NA	NA	HQ450557	Bodotriidae	Cyclaspis sp.	HQ450557	
Bod06	UoB	KV-09 2011	Bio material=146	MK613886.1	Bodotriidae	Iphinoe serrata	seq4	
Cer01	IceAGE	1057	DZMB-HH-68388	MZ402679	Ceratocumatidae	Cimmerius reticulatus	ICE1-Cer001	
Cer01	IceAGE	1072	DZMB-HH-68362	MZ402678	Ceratocumatidae	Cimmerius reticulatus	ICE1-Cer002	
Cer01	IceAGE	1072	DZMB-HH-68349	MZ402677	Ceratocumatidae	Cimmerius reticulatus	ICE1-Cer003	
Dia01	UoB	09.01.28-2	Bio material=160409-8	MK613898.1	Diastylidae	Diastylis cornuta	seq25	
Dia01	UoB	VGPT1-22 2009	Bio material=9-13	MK613897.1	Diastylidae	Diastylis cornuta	seq26	
Dia03	UoB	UNIS 2007-129	Bio material=031109-12	MK613904.1	Diastylidae	Diastylis goodsiri	seq31	
Dia04	UoB	BS 14-19	Bio material=21-22	MK613901.1	Diastylidae	Diastylis laevis	seq33	
Dia05	UoB	BS 14-19	Bio material=26-28	MK613911.1	Diastylidae	Diastylis lucifera	seq35	
Dia06	IceAGE	1144	DZMB-HH-68295	MZ402685	Diastylidae	Diastylis polaris	ICE1-Dia010	
Dia06	IceAGE	1144	DZMB-HH-68297	MZ402686	Diastylidae	Diastylis polaris	ICE1-Dia016	
Dia06	IceAGE	1184	DZMB-HH-68262	MZ402683	Diastylidae	Diastylis polaris	ICE1-Dia003	
Dia06	IceAGE	1184	DZMB-HH-68263	MZ402682	Diastylidae	Diastylis polaris	ICE1-Dia006	
Dia06	IceAGE	1184	DZMB-HH-68234	MZ402684	Diastylidae	Diastylis polaris	ICE1-Dia009	
Dia06	IceAGE	1191	DZMB-HH-68259	MZ402687	Diastylidae	Diastylis polaris	ICE1-Dia019	
Dia06	UoB	H2DEEP-RP-1	Bio material=29-31	MK613902.1	Diastylidae	Diastylis polaris	seq39	
Dia06	MAREANO	R488-379, BT	Bio material=32-33	MK613903.1	Diastylidae	Diastylis polaris	seq40	
Dia07	UoB	UNIS 2009-73	Bio material=031109-19	MK613905.1	Diastylidae	Diastylis rathkei	seq36	
NA	NA	NA	NA	HQ450555	Diastylidae	Diastylis rathkei	HQ450555	
NA	NA	NA	NA	U81512	Diastylidae	Diastylis sculpta	U81512	
Dia08	UoB	UNIS 2007-140	Bio material=031109-16	MK613906.1	Diastylidae	Diastylis cf. spinulosa	seq38	
Dia09	UoB	11.05.15-1	Bio material=35-37	MK613899.1	Diastylidae	Diastylis tumida	seq42	
Dia09	MAREANO	R721-126, RP	Bio material=152-153	MK613900.1	Diastylidae	Diastylis tumida	seq43	
Dia10	IceAGE	983	DZMB-HH-68413	MZ402689	Diastylidae	Diastyloides atlanticus	ICE1-Dia011	
Dia10	IceAGE	983	DZMB-HH-68434	MZ402688	Diastylidae	Diastyloides atlanticus	ICE1-Dia024	
Dia11	UoB	11.03.11-2	Bio material=D4-D6	MK613910.1	Diastylidae	Diastyloides biplicatus	seq44	
Dia12	UoB	11.05.10-3	Bio material=38-40	MK613907.1	Diastylidae	Diastyloides serratus	seq47	
Dia12	UoB	11.05.11-2C	Bio material=159, 174	MK613909.1	Diastylidae	Diastyloides serratus	seq48	
Dia12	UoB	BS 82-147	Bio material=1004-1006	MK613908.1	Diastylidae	Diastyloides serratus	seq49	
NA	NA	NA	NA	HQ450556	Diastylidae	Diastylopsis sp.	HQ450556	
Dia14	IceAGE	1123	DZMB-HH-68456	MZ402704	Diastylidae	Leptostylis ampullacea	ICE1-Dia018	
Dia14	IceAGE	1123	DZMB-HH-68443	MZ402711	Diastylidae	Leptostylis ampullacea	ICE1-Dia001	
Dia14	IceAGE	1123	DZMB-HH-68445	MZ402708	Diastylidae	Leptostylis ampullacea	ICE1-Dia007	
Dia14	IceAGE	1136	DZMB-HH-68266	MZ402710	Diastylidae	Leptostylis ampullacea	ICE1-Dia002	
Dia14	IceAGE	1136	DZMB-HH-68267	MZ402709	Diastylidae	Leptostylis ampullacea	ICE1-Dia005	
Dia14	IceAGE	1136	DZMB-HH-68268	MZ402707	Diastylidae	Leptostylis ampullacea	ICE1-Dia008	
Dia14	IceAGE	1144	DZMB-HH-68296	MZ402706	Diastylidae	Leptostylis ampullacea	ICE1-Dia013	
Dia14	IceAGE	1191	DZMB-HH-68258	MZ402705	Diastylidae	Leptostylis ampullacea	ICE1-Dia014	
Dia15	IceAGE	1136	DZMB-HH-68269	MZ402712	Diastylidae	Leptostylis borealis	ICE1-Dia015	
Dia15	IceAGE	1219	DZMB-HH-68403	MZ402713	Diastylidae	Leptostylis borealis	ICE1-Dia017	
Dia16-A	UoB	11.05.10-1	Bio material=41-43	MK613921.1	Diastylidae	Leptostylis longimana	seq52	
Dia16-A	UoB	BS 82-147	Bio material=49-50	MK613922.1	Diastylidae	Leptostylis longimana	seq53	
Dia16-B	PASCAL	24/5	DZMB-HH-63369	MZ402723	Diastylidae	Leptostylis cf. longimana	P-Dias001	
Dia16-B	PASCAL	24/5	DZMB-HH-63370	MZ402714	Diastylidae	Leptostylis cf. longimana	P-Dias002	
Dia16-B	PASCAL	24/5	DZMB-HH-59943	MZ402722	Diastylidae	Leptostylis cf. longimana	P-Dias028	
Dia16-B	PASCAL	24/5	DZMB-HH-63371	MZ402717	Diastylidae	Leptostylis cf. longimana	P-Dias032	
Dia16-B	PASCAL	25/5	DZMB-HH-59218	MZ402718	Diastylidae	Leptostylis cf. longimana	P-Dias007	
Dia16-B	PASCAL	30/1	DZMB-HH-63337	MZ402716	Diastylidae	Leptostylis cf. longimana	P-Dias003	
Dia16-B	PASCAL	30/1	DZMB-HH-59533	MZ402719	Diastylidae	Leptostylis cf. longimana	P-Dias027	
Dia16-B	PASCAL	30/1	DZMB-HH-63343	MZ402720	Diastylidae	Leptostylis cf. longimana	P-Dias031	
Dia16-B	PASCAL	32/3	DZMB-HH-63330	MZ402721	Diastylidae	Leptostylis cf. longimana	P-Dias004	
Dia16-B	PASCAL	32/3	DZMB-HH-63331	MZ402724	Diastylidae	Leptostylis cf. longimana	P-Dias029	
Dia16-B	PASCAL	32/3	DZMB-HH-63334	MZ402715	Diastylidae	Leptostylis cf. longimana	P-Dias030	
Dia17	IceAGE	983	DZMB-HH-68414	MZ402702	Diastylidae	Leptostylis sp. 1	ICE1-Dia012	
Dia17	IceAGE	983	DZMB-HH-68418	MZ402701	Diastylidae	Leptostylis sp. 1	ICE1-Dia025	
Lam01	Alaska	90626	Bio material=200912-9	MK613925.1	Lampropidae	Alamprops augustinensis	seq87	
Lam02	IceAGE	983	DZMB-HH-68421	MZ402676	Lampropidae	Chalarostylis elegans	ICE1-Lam009	
Lam02	IceAGE	983	DZMB-HH-68424	MZ402675	Lampropidae	Chalarostylis elegans	ICE1-Lam017	
Lam04	UoB	BIOICE3669	Bio material=187-188, ma6	MK613924.1	Lampropidae	Hemilamprops assimilis	seq80	
Lam05-A	UoB	BS 86-151	Bio material=63-64	MK613913.1	Lampropidae	Hemilamprops cristatus	seq81	
Lam05-A	UoB	BS 86-151	Bio material=65	MK613914.1	Lampropidae	Hemilamprops cristatus	seq82	
Lam05-B	IceAGE	983	DZMB-HH-68420	MZ402695	Lampropidae	Hemilamprops cf. cristatus	ICE1-Lam002	
Lam05-B	IceAGE	983	DZMB-HH-68436	MZ402696	Lampropidae	Hemilamprops cf. cristatus	ICE1-Lam008	
Lam05-A	IceAGE	1123	DZMB-HH-68446	MZ402697	Lampropidae	Hemilamprops cf. cristatus	ICE1-Lam018	
Lam06	IceAGE	983	DZMB-HH-68419	MZ402692	Lampropidae	Hemilamprops cf. diversus	ICE1-Lam001	
Lam06	IceAGE	983	DZMB-HH-68435	MZ402693	Lampropidae	Hemilamprops cf. diversus	ICE1-Lam006	
Lam06	IceAGE	983	DZMB-HH-68422	MZ402694	Lampropidae	Hemilamprops cf. diversus	ICE1-Lam010	
Lam06	IceAGE	983	DZMB-HH-68423	MZ402691	Lampropidae	Hemilamprops cf. diversus	ICE1-Lam011	
Lam07	IceAGE	1072	DZMB-HH-68363	MZ402698	Lampropidae	Hemilamprops pterini	ICE1-Lam005	
Lam07	IceAGE	1072	DZMB-HH-68364	MZ402699	Lampropidae	Hemilamprops pterini	ICE1-Lam013	
Lam08	UoB	BS 28-44	Bio material=66-67	MK613923.1	Lampropidae	Hemilamprops roseus	seq83	
Lam10	IceAGE	1072	DZMB-HH-68365	MZ402690	Lampropidae	Hemilamprops sp. 2	ICE1-Lam015	
Lam11	IceAGE	1136	DZMB-HH-68270	MZ402700	Lampropidae	Hemilamprops uniplicatus	ICE1-Lam003	
Lam11	UoB	11.05.15-1	Bio material=68-70	MK613915.1	Lampropidae	Hemilamprops uniplicatus	seq84	
Lam11	UoB	SFND-08R 2011	Bio material=71-72	MK613916.1	Lampropidae	Hemilamprops uniplicatus	seq85	
Lam12	UoB	VGPT1-22 2009	Bio material=61-62	MK613917.1	Lampropidae	Mesolamprops denticulatus	seq88	
Lam13	IceAGE	1072	DZMB-HH-68366	MZ402737	Lampropidae	Platysympus typicus	ICE1-Lam016	
Lam13	IceAGE	1136	DZMB-HH-68271	MZ402736	Lampropidae	Platysympus typicus	ICE1-Lam004	
Lam13	UoB	UNIS 2009-71	Bio material=031109-15	MK613918.1	Lampropidae	Platysympus typicus	seq89	
Lam13	MAREANO	R814-22, RP	Bio material=ma14	MK613919.1	Lampropidae	Platysympus typicus	seq90	
Leu01	UoB	BS 75-135	Bio material=79-82	MK613870.1	Leuconidae	Eudorella emarginata	seq59	
Leu02	UoB	BS 34-56	Bio material=88-93	MK613887.1	Leuconidae	Eudorella hirsuta	seq62	
Leu02	UoB	11.03.09-1	Bio material=94-96, 102	MK613888.1	Leuconidae	Eudorella hirsuta	seq63	
NA	NA	NA	NA	U81513	Leuconidae	Eudorella pusilla	U81513	
Leu04-A	UoB	BS 28-44	Bio material=97, 99	MK613881.1	Leuconidae	Eudorella truncatula	seq64	
Leu04-A	UoB	BS 75-135	Bio material=200	MK613882.1	Leuconidae	Eudorella truncatula	seq67	
Leu04-B	UoB	11.01.19.1	Bio material=100-101	MK613884.1	Leuconidae	Eudorella truncatula	seq65	
Leu04-B	UoB	11.01.21-1	Bio material=1007-1008	MK613883.1	Leuconidae	Eudorella truncatula	seq68	
Leu04-C	MAREANO	R754-132, RP	Bio material=ma5	MK613885.1	Leuconidae	Eudorella truncatula	seq69	
Leu05	IceAGE	1123	DZMB-HH-68457	MZ402728	Leuconidae	Leucon (Alytoleucon) pallidus	ICE1-Leu019	
Leu05	IceAGE	1136	DZMB-HH-68274	MZ402729	Leuconidae	Leucon (Alytoleucon) pallidus	ICE1-Leu002	
Leu05	IceAGE	1136	DZMB-HH-68276	MZ402731	Leuconidae	Leucon (Alytoleucon) pallidus	ICE1-Leu005	
Leu05	IceAGE	1136	DZMB-HH-68278	NA	Leuconidae	Leucon (Alytoleucon) pallidus	ICE1-Leu008	
Leu05	IceAGE	1144	DZMB-HH-68298	MZ402730	Leuconidae	Leucon (Alytoleucon) pallidus	ICE1-Leu003	
Leu05	IceAGE	1144	DZMB-HH-68299	MZ402725	Leuconidae	Leucon (Alytoleucon) pallidus	ICE1-Leu006	
Leu05	IceAGE	1144	DZMB-HH-68300	MZ402726	Leuconidae	Leucon (Alytoleucon) pallidus	ICE1-Leu009	
Leu05	IceAGE	1219	DZMB-HH-68404	MZ402727	Leuconidae	Leucon (Alytoleucon) pallidus	ICE1-Leu010	
Leu05	UoB	BS 82-147	Bio material=1001-1003	MK613892.1	Leuconidae	Leucon (Alytoleucon) pallidus	seq77	
Leu05	UoB	11.01.19.1	Bio material=117-119	MK613891.1	Leuconidae	Leucon (Alytoleucon) pallidus	seq78	
NA	NA	NA	NA	HQ450522	Leuconidae	Leucon (Crymoleucon) antarcticus	HQ450522	
NA	NA	NA	NA	HQ450543	Leuconidae	Leucon (Crymoleucon) intermedius	HQ450543	
NA	NA	NA	NA	HQ450549	Leuconidae	Leucon (Crymoleucon) intermedius	HQ450549	
NA	NA	NA	NA	HQ450550	Leuconidae	Leucon (Crymoleucon) intermedius	HQ450550	
NA	NA	NA	NA	HQ450537	Leuconidae	Leucon (Crymoleucon) rossi	HQ450537	
Leu07	UoB	BS 22-32	Bio material=103-108	MK613889.1	Leuconidae	Leucon (Leucon) acutirostris	seq70	
NA	NA	NA	NA	HQ450551	Leuconidae	Leucon (Leucon) assimilis	HQ450551	
NA	NA	NA	NA	HQ450552	Leuconidae	Leucon (Leucon) assimilis	HQ450552	
NA	NA	NA	NA	HQ450553	Leuconidae	Leucon (Leucon) assimilis	HQ450553	
Leu08	UoB	09.01.28-2	Bio material=109-111	MK613895.1	Leuconidae	Leucon (Leucon) nathorsti	seq72	
Leu08	UoB	BS 75-135	Bio material=112-114	MK613893.1	Leuconidae	Leucon (Leucon) nathorsti	seq73	
Leu09	UoB	UNIS 2009-27	Bio material=031109-9	MK613894.1	Leuconidae	Leucon (Leucon) aff. nathorsti	seq75	
Leu10	UoB	UNIS2009-4	Bio material=115-116	MK613890.1	Leuconidae	Leucon (Leucon) nasicoides	seq74	
Leu11	IceAGE	1136	DZMB-HH-68273	MZ402734	Leuconidae	Leucon (Leucon) profundus	ICE1-Leu001	
Leu11	IceAGE	1136	DZMB-HH-68275	MZ402732	Leuconidae	Leucon (Leucon) profundus	ICE1-Leu004	
Leu11	IceAGE	1136	DZMB-HH-68277	MZ402733	Leuconidae	Leucon (Leucon) profundus	ICE1-Leu007	
Leu14	IceAGE	1123	DZMB-HH-68449	MZ402735	Leuconidae	Leucon (Macrauloleucon) spinulosus	ICE1-Leu018	
NA	NA	NA	NA	HQ450554	Leuconidae	Leucon sp.	HQ450554	
Nan03	UoB	BS 86-151	Bio material=120-122	MK613876.1	Nannastacidae	Campylaspis costata	seq6	
Nan04	UoB	BS 34-56	Bio material=1-3	MK613874.1	Nannastacidae	Campylaspis globosa	seq9	
Nan04	IceAGE	1057	DZMB-HH-68390	MZ402662	Nannastacidae	Campylaspis cf. globosa	ICE1-Nann014	
Nan05	IceAGE	1057	DZMB-HH-68389	MZ402663	Nannastacidae	Campylaspis horrida	ICE1-Nann013	
Nan05	MAREANO	R721-126, RP	Bio material=123	MK613877.1	Nannastacidae	Campylaspis horrida	seq10	
Nan06	PASCAL	24/5	DZMB-HH-63414	MZ402664	Nannastacidae	Campylaspis intermedia	P-Nann009	
Nan07	PASCAL	24/5	DZMB-HH-63399	MZ402666	Nannastacidae	Campylaspis rubicunda	P-Nann001	
Nan07	PASCAL	24/5	DZMB-HH-63400	MZ402665	Nannastacidae	Campylaspis rubicunda	P-Nann002	
Nan07	PASCAL	24/5	DZMB-HH-63401	MZ402670	Nannastacidae	Campylaspis rubicunda	P-Nann003	
Nan07	PASCAL	24/5	DZMB-HH-63402	MZ402669	Nannastacidae	Campylaspis rubicunda	P-Nann004	
Nan07	PASCAL	24/5	DZMB-HH-63405	MZ402667	Nannastacidae	Campylaspis rubicunda	P-Nann006	
Nan07	PASCAL	24/5	DZMB-HH-59833	MZ402668	Nannastacidae	Campylaspis rubicunda	P-Nann011	
Nan07	PASCAL	24/5	DZMB-HH-63357	MZ402671	Nannastacidae	Campylaspis rubicunda	P-Nann012	
Nan09	IceAGE	1072	DZMB-HH-68369	MZ402661	Nannastacidae	Campylaspis sp. 2	ICE1-Nann005	
Nan10	IceAGE	1136	DZMB-HH-68280	MZ402672	Nannastacidae	Campylaspis sulcata	ICE1-Nann002	
Nan10	IceAGE	1136	DZMB-HH-68281	MZ402674	Nannastacidae	Campylaspis sulcata	ICE1-Nann004	
Nan10	IceAGE	1136	DZMB-HH-68282	MZ402673	Nannastacidae	Campylaspis sulcata	ICE1-Nann006	
Nan10	UoB	11.05.15-1	Bio material=134-139	MK613875.1	Nannastacidae	Campylaspis sulcata	seq14	
Nan11	UoB	11.05.15-1	Bio material=140-145	MK613878.1	Nannastacidae	Campylaspis undata	seq21	
Nan18	IceAGE	1072	DZMB-HH-68370	MZ402738	Nannastacidae	Styloptocuma gracillimum	ICE1-Nann008	
Pse01	UoB	UNIS 2009-36	Bio material=156-158	MK613871.1	Pseudocumatidae	Petalosarsia declivis	seq5	
NA	NA	NA	NA	AJ388110	Tanaidacea	Apseudopsis latreillii	AJ388110	
NA	NA	NA	NA	DQ305106	Isopoda	Asellus (Asellus) aquaticus	DQ305106	
NA	NA	NA	NA	AF260869	Isopoda	Colubotelson thompsoni	AF260869	
NA	NA	NA	NA	AF260870	Isopoda	Crenoicus buntiae	AF260870	
NA	NA	NA	NA	AY693421	Isopoda	Haploniscus sp.	AY693421	
NA	NA	NA	NA	AF259533	Isopoda	Paramphisopus palustris	AF259533	
NA	NA	NA	NA	DQ305111	Isopoda	Proasellus remyi remyi	DQ305111	
NA	NA	NA	NA	MK813124	Amphipoda	Amphipoda sp.	MK813124	

Phylogenetic analyses

Raw sequences were assembled and manually curated in Geneious® version 9.8.1 (Kearse et al., 2012). Consensus sequences were generated and blasted against GenBank database to identify potential contaminant sequences (e.g., bacterial sequences). We further included 67 cumacean sequences published on GenBank, 51 sequences originating from North Atlantic cumaceans currently studied at the University Bergen as well as 16 from outside the study area (Table 2).

Due to the large number of substitutions and indels, the alignment of all sequenced species included many long, ambiguously aligned regions, which would compromise the following analyses. For this reason, we split the data into four subsets of more closely related (and thus more similar) sequences, based on morphological family taxa and a preliminary phylogenetic analysis on the complete dataset (dataset 1 in Fig. 2; Table 3). These family-based alignments had fewer ambiguities and gaps and were thus used for subsequent analyses. Alignments were calculated separately for each of these four subsets with MAFFT 7.402 (Katoh, 2002; Katoh & Standley, 2013) on the CIPRES Science Gateway version 3.3 (Miller, Pfeiffer & Schwartz, 2010) using the L-INS-I algorithm and subsequently trimmed manually in BioEdit© version 7.0.5.3 (Hall, 1999). One outgroup species (represented by a member of one of the respective other cumacean families) was included in the alignments for the phylogenetic analyses but removed to further improve the alignment for genetic distance analyses.

Figure 2 Dataset 1-Phylogenetic analyses based on the 16S rRNA gene region of Cumacea from Northern Atlantic to Arctic waters.

Included in the Bayesian analysis were 80 sequences of all morphologically determined cumacean species, 67 sequences published on GenBank of which 51 sequences are originating from North Atlantic cumaceans currently studied at the University Bergen as well as 16 from outside the study area and eight other peracarid outgroup sequences (Isopoda, Amphipoda, Tanaidacea). The branch labels indicate posterior probability scores in percent decimal values (1 = absolute support in all calculated trees). The scale bar at the bottom of the tree shows nucleotide substitutions per site of sequence. Colours are representing family taxa, which were split into dataset II–V for subsequent genetic distance analyses. Genetic lineages are separated by assigned letters A/B/C.

Figure 3 Dataset 2-Phylogenetic relationships inferred by Bayesian analysis of the Leuconidae.

The branch labels indicate posterior probability scores. The scale bar at the bottom of the tree shows nucleotide substitutions per site of sequence. Sequences in parentheses were excluded from genetic distance analyses due to insufficient sequence length. Vertical bars indicate the species delimited based on morphology, ABGD and GMYC with colors representing genera. Genetic lineages are separated by assigned letters A/B/C.

Table 3 Summary of datasets 1–5.

Number of sequences and the resulting alignment length in base pairs integrated in the Bayesian phylogenetic analyses.

Dataset	Taxa	Sequences for topology (n)	Outgroup sequences (n)	Alignment length (bp)	
1	All Cumacea cumulative set	155	8	598	
2	Leuconidae	38	1	525	
3	Bodotriidae and Nannastacidae	31	1	524	
4	Diastylidae and Pseudocumatidae	53	1	549	
5	Ceratocumatidae and Lampropidae	29	1	525	

The best-fitting evolutionary model was identified in MEGA X (Kumar et al., 2018), resulting in the General Time Reversible Model with invariable sites and gamma distribution (GTR + G + I; Lanave et al., 1984; Rodriguez et al., 1990; Nylander et al., 2004) for all data sets. Subsequent phylogenetic analyses were performed with MrBayes version 3.2.7a (Ronquist et al., 2012) with ‘nruns = 4’ and ‘nchains = 6’, 50 × 106 generations and a sample frequency of 5,000. The first 25% of sampled trees were discarded as burn-in. The resulting consensus trees were visualized with FigTree version 1.4.4 (Rambaut, 2018a).

Two different species delimitation methods were employed, one tree-based (general mixed Yule coalescent, GMYC, Pons et al., 2006) and one distance-based (automatic barcode gap discovery, ABGD, Puillandre et al., 2012). The single threshold model of GMYC was performed in R (R Core Team) for each of the four subsets. The required ultrametric trees were calculated with BEAST 2.5 (Bouckaert et al., 2019), employing the Yule coalescent prior, enforcing the ingroup as monophyletic and running the analyses for 30 * 106 generations and sampling every 3,000th generation. Convergence was assessed in Tracer (Rambaut et al., 2018b) and the final tree produced with TreeAnnotator (Bouckaert et al., 2019), removing the first 25% of generations as burn-in.

The relatively large number of potential singleton taxa could be problematic for tree-based species delimitation approaches. For ABGD, uncorrected p-distances were precomputed without an evolutionary substitution model in MEGA X including transitions and transversions as substitution mutations and missing data was treated by pairwise deletion. Sequences shorter than 300 bp were excluded. The web-based version of ABGD was used (https://bioinfo.mnhn.fr/abi/public/abgd/abgdweb.html), setting Pmin = 0.01, Pmax = 0.1, the relative gap width (X) to 0.5 and the number of steps to 100.

Distribution maps

For a general occurrence range overview, distribution maps were created using the software R version 3.5.3 and the PlotSvalbard package version 0.8.5 (Vihtakari, 2019) for each species. A single map incorporates the available georeferenced records from the open-access portal OBIS (OBIS, 2019; https://mapper.obis.org/, accessed 19/09/2019) and either the type locality or reference localities from earlier publications. Additionally, new occurrence records, not integrated in the OBIS platform, were added from published literature (e.g., Watling & Gerken, 2005) as well as records from other publicly accessible occurrence record libraries (e.g., MAREANO platform, Table S3). The new OBIS dataset Icelandic Cumacea (ICECU) was created for IceAGE and PASCAL specimens (Uhlir et al., 2021; http://ipt.vliz.be/eurobis/resource?r=cumacea_pascal_iceage). Information on cumacean species sampled and identified by the MAREANO project are accessible for specific taxa via the species-list portal (http://webprod1.nodc.no:8080/marbunn_web/viewspecies).

Results

Combined approach: morphological and molecular species delimitation

The 947 investigated specimens were assigned to 77 morphological species, representing all seven known families (Table S4). For 58 species, identification to a known species taxon was possible. In all other 19 cases, specimens were assigned to genus or family level, but clearly differed morphologically from all other species of these genera or families identified in our study. The largest number of species were assigned to the Nannastacidae (20 species), followed by the Diastylidae (19), the Lampropidae (14) and the Leuconidae (15). In terms of DNA quality and success rate, the Marine Animal Tissue Genomic DNA kit yielded the best results for the cumaceans and can, thus, be recommended for further studies on this taxon.

In total, 131 specimens were included in the genetic analyses, representing 54 of the 77 morphologically identified species (Table 2). The Bayesian phylogenetic analysis of dataset 1 in Fig. 2 (all taxa) resulted in the monophyly of the Cumacea (posterior probability (pp) = 1). Except for the Nannastacidae (pp = 1), families were not recovered as monophyletic (Figs. 3–6), but this was not surprising, as a single fastly evolving marker like 16S is not suitable to properly resolve such deep nodes.

Figure 4 Dataset 3-Phylogenetic relationships inferred by Bayesian analysis of the Bodotriidae and Nannastacidae.

The branch labels indicate posterior probability scores. The scale bar at the bottom of the tree shows nucleotide substitutions per site of sequence. Sequences in parentheses were excluded from genetic distance analyses due to insufficient sequence length. Vertical bars indicate the species delimited based on morphology, ABGD and GMYC with colors representing genera. Genetic lineages are separated by assigned letters A/B.

Figure 5 Dataset 4-Phylogenetic relationships inferred by Bayesian analysis of the Diastylidae and the Pseudocumatidae.

The branch labels indicate posterior probability scores. The scale bar at the bottom of the tree shows nucleotide substitutions per site of sequence. Sequences in parentheses were excluded from genetic distance analyses due to insufficient sequence length. Vertical bars indicate the species delimited based on morphology, ABGD and GMYC with colors representing genera. Connecting line in Dia14, -16-A, -16-B indicates the same putative species based on morphological identification. Genetic lineages are separated by assigned letters A/B.

Figure 6 Dataset 5-Phylogenetic relationships inferred by Bayesian analysis of Ceratocumatidae and Lampropidae.

The branch labels indicate posterior probability scores. The scale bar at the bottom of the tree shows nucleotide substitutions per site of sequence. Sequences in parentheses were excluded from genetic distance analyses due to insufficient sequence length. Vertical bars indicate the species delimited based on morphology, ABGD and GMYC with colors representing genera. Connecting line in Lam05 indicates the same putative species based on morphological identification. Genetic lineages are separated by assigned letters A/B.

The ABGD analyses delimited 53 genetic lineages (representing putative species). With few exceptions, lineages were separated by a clear barcoding gap, with the vast majority of intra-lineage distances being <1% and inter-lineage distances > 8% (mostly 15–45%) (Fig. S1; Tables S5–S12). Cases of intra-lineage p-distance exceeding 1% were Diastylis rathkei Krøyer, 1841 (Dia07, 4%; Tables S5, S6) and Hemilamprops cristatus G. O. Sars, 1870 (Lam05-A, 2%; Tables S7, S8). GMYC resulted in nearly identical species delimitation, only Diastyloides biplicatus G. O. Sars, 1865 (Dia11) and Diastyloides serratus G. O. Sars, 1865 (Dia12) were grouped together (because these are separated by a genetic distance of 18%, we consider this to be an artifact due to the high number of singletons (D. biplicatus is also singleton) and treat them separately in the following as suggested by AGBD).

Inconsistencies between morphological and molecular species delimitation occurred in eight cases, which are summarized in Table 4. In four cases, genetic divergence was higher than expected by prior morphological determination suggesting cryptic diversity. Cyclaspis longicaudata Sars, 1865 was split into two distinct lineages (Bod05, A–B; Table S9, S10) as were Leptostylis borealis Stappers, 1908 (Dia15, A–B; Tables S5, S6) and Leptostylis sp. 1 (Dia17, A–B; Tables S5, S6). Eudorella truncatula Bate, 1856 was split into three lineages (Leu04, A–C; Tables S11, S12). Conversely, Leucon (Leucon) aff. nathorsti Ohlin, 1901 (Leu09) and L. (L.) nathorsti (Leu08) were treated as two morphologically differing species based on prior determination following Hansen (1920) (Leu09 with rather pointy rostrum; two dorsolateral teeth on frontal lobe). However, the low genetic distance (1%) suggests that they belong to the same lineage (Tables S11, S12). Finally, two problematic cases highlighted the mismatch between morphological and genetic delimitation. First, Leptostylis longimana Sars, 1865 was genetically split into two morphologically cryptic lineages (Dia16, A–B; Tables S5, S6), of which Dia16-A was collected close to the species’ type locality on the continental shelf and a Norwegian fjord and Dia16-B from Arctic Polar Water (APW). Furthermore, Dia16-B was genetically identical to Leptostylis ampullacea Lilljeborg, 1855 (Dia14), which was collected in Icelandic waters (Norwegian Sea Arctic Intermediate Water, APW-NSAIW) more than 2,500 km further north. However, after re-examination these specimens could barely be distinguished based only on weakly discriminating morphological characters following G. O. Sars (1900; clumsier form of body) from specimens identified as the original Leptostylis longimana (Dia16-A). Second, one species (Hemilamprops cf. cristatus; Lam05-B) morphologically very closely resembled Hemilamprops cristatus (Lam05-A) but was eventually differentiated based on a shorter rostrum and smaller, but more teeth within the serrated dorsal crest in Lam05-A. Also, genetic analyses suggested two lineages with strong divergence (23% p-distance), however, the Lam05-A specimen (sequence ID ICE1-Lam018) from the Greenland slope in Subarctic waters (APW-NSAIW), morphologically assigned to Lam05-B, clustered together with Lam05-A from the Norwegian continental shelf. All other Lam06 specimens were from Iceland Sea Overflow Water (ISOW) in the Iceland Basin.

Table 4 Summary of taxonomic incongruences, morphological variability, and potential cryptic diversity cases.

Species ID	Putative species	Sequence ID (Field ID)	Region	Depth range (m)	Water mass	
Taxonomic incongruences	
Dia06
(Fig. 11C)	Diastylis polaris
aka
’Diastylis stygia’	seq39	Jan Mayen (Norway)	2,542	Arctic, Subarctic	
seq40	Eggakanten (Norway)	2,241	Subarctic	
ICE1-Dia003, ICE1-Dia006, ICE1-Dia009	Norwegian Sea Basin	1,819	Subarctic	
ICE1-Dia010, ICE1-Dia016	East Greenland Denmark Strait	1,281	Arctic, Subarctic	
ICE1-Dia019	North-East Iceland Norwegian Sea	1,574	Subarctic	
Lam13
(Fig. 11E)	Platysympus typicus
aka
’Platysympus tricarinatus’	ICE1-Lam004	East Greenland Denmark Strait	315	Arctic, Subarctic	
ICE1-Lam016	South Iceland Irminger Basin	1,593	North Atlantic	
seq89	Svalbard	497	Arctic	
seq90	Continental Shelf (Norway)	224	North Atlantic	
Morphological variability	
Dia14
(Fig. 11B)
&
Dia16-B
(Fig. 11A)	Leptostylis ampullacea	ICE1-Dia018, ICE1-Dia002, ICE1-Dia005, ICE1-Dia008, ICE1-Dia001, ICE1-Dia004, ICE1-Dia007, ICE1-Dia013, ICE1-Dia014	North East Iceland	700–1,500	Arctic, Subarctic	
Leptostylis cf. longimana B	P-Dias001, P-Dias002, P-Dias028, P-Dias032, P-Dias007, P-Dias027, P-Dias031, P-Dias003, P-Dias029, P-Dias030	Yermak-Plateau (Svalbard)	700–1,500	Arctic	
Leu08
& Leu09
(Fig. S2W)	Leucon nathorsti	seq72, seq 73	Fanafjorden, Skagerak	180–246	North Atlantic	
Leucon aff. nathorsti	seq75	Svalbard	56	Subarctic	
Morphological & molecular data: cryptic diversity	
Leu04-A/-B/-C
(Fig. S2U)	Eudorella truncatula A	seq64, seq67	Skagerak	250	North Atlantic	
Eudorella truncatula B	seq65, seq68	Fensfjorden, Hjeltefjorden	200–400	North Atlantic	
Eudorella truncatula C	seq69	Northern Norway	800	Subarctic	
Bod05-A/-B
(Fig. 13E)	Cyclaspis longicaudata A	seq2, seq3	Hjeltefjorden	240–330	North Atlantic	
Cyclaspis longicaudata B	ICE1-Bod001, ICE1-Bod002	South Iceland Basin	2,500	North Atlantic	
Dia15-A/-B
(Fig. 12A)	Leptostylis borealis A	ICE1-Dia015	Greenland shelf	300	Subarctic	
Leptostylis borealis B	ICE1-Dia017	North East Iceland	500	Subarctic	
Dia17-A/-B
(Fig. S2L)	Leptostylis sp. 1 A	ICE1-Dia012	South Iceland Basin	2,500	North Atlantic	
Leptostylis sp. 1 B	ICE1-Dia025	South Iceland Basin	2,500	North Atlantic	
Dia16-A/-B
(Fig. 11A)	Leptostylis cf. longimana B	P-Dias001, P-Dias002, P-Dias028, P-Dias032, P-Dias007, P-Dias027, P-Dias031, P-Dias003, P-Dias029, P-Dias030	Yermak-Plateau (Svalbard)	700–1,500	Arctic, Subarctic	
Leptostylis longimana A	seq52, seq53	Sognesjøen & Skagerak	500	North Atlantic	
Lam05-A/-B
(Fig. 11F)	Hemilamprops cristatus/Hemilamprops cf. cristatus	seq81, seq82, ICE1-Lam018	Skagerak & Greenland slope	700	North Atlantic, Subarctic	
Lam05-B
(Fig. 11F)	Hemilamprops cf. cristatus	ICE1-Lam002, ICE1-Lam008	South Iceland Basin	2,500	North Atlantic	

Biogeographical data mining in OBIS

Within the investigated area, a total of 11,714 occurrence records including 44,933 individual specimens were extracted from OBIS (9,151 records), literature and other databases (2,270), and the new ICECU dataset added 293 records (Fig. 7). Out of these, about 6,200 records are in shelf regions up to 250 m depth, about 3,900 records in shelf-break regions between 250–1,000 m and 639 records below 1,000 m in the deep sea, excluding about 780 records with no available depth information. More than half of the specimens (25,496) were classified on order level as ‘Cumacea indet.’, whereas 19,437 specimens were identified to family or a lower taxonomic level. In total, 109 known species are recorded, of which 18 species of five families were recorded for the first time on the OBIS platform within the ecoregions 1, 2, 4, 5 and 7 (Fig. 8). The amount of data and specimen records varied remarkably among the predefined ecoregions 1–8. Ecoregion 4 was assigned to North Atlantic water mass characteristics comprising the highest specimen count (20,101), followed by ecoregion 5 (11,664), 2 (7,667) and 7 (3,505), which are composed of a mixture of North Atlantic and Arctic water masses and were, thus, assigned to Subarctic water-mass characteristics. Arctic water-mass ecoregions 8 (858 specimens), 1 (232), 6 (490) and 3 (29) contributed the lowest specimen sampling effort. There was a general trend of decreasing number of taxa (species diversity) with fewer specimens following the northern extension of the North Atlantic Current (NAC). For example, out of 232 individual specimens in ecoregion 1, 203 were assigned to 30 different species, while ecoregion 8, one of the last ecoregions influenced by the NAC, had a higher sampling effort with 858 specimens, but a lower species diversity with 407 individuals assigned to 18 species.

Figure 7 Summarized occurrence records of ‘Cumacea’ and their taxonomic level of determination.

(A) Occurrence data of present (OBIS) and newly added records (MAREANO, ICECU, literature) summarized and separated into the predefined marine ecoregions (1–8) and water masses (red: North Atlantic; black: Sub-Arctic; blue: Arctic). Bar plots show the total number of specimens and their taxonomic level of determination (see legend) in relation to the total number of determined taxa. (B) Surface current branching of the North Atlantic current (4) entering the Arctic Ocean via the Norwegian Sea (5) up to Arctic water masses in the Kara Sea (8) and the outflow off the Greenland coast (3) passing the Denmark Strait (2) off Iceland (modified from Townsend, 2012).

Figure 8 Occurrence records within defined ecoregions for the order Cumacea publicly accessible and additional records contributed by the new ICECU dataset.

Overview map for the order Cumacea of present occurrence records (blue), which are publicly accessible on the OBIS platform, and new records (orange), which were contributed by the MAREANO platform, the ICECU dataset and literature data within predefined water masses (red, North Atlantic; black, Sub-Arctic; cyan, Arctic) and ecoregions (1, Arctic Basin; 2, East Greenland Sea; 3, North Greenland Sea; 4, North Atlantic Ocean; 5, Norwegian Sea; 6, White Sea; 7, Barents Sea; 8, Kara Sea) after Curtis (1975), Hansen & Østerhus (2000) and Schlichtholz & Houssais (2002). Main focus of this study is on specimens from the bold highlighted ecoregions 1, 2 and 4.

Species distribution patterns within ecoregions

Overall, the composition of taxa was observed to change from a Northern Atlantic-boreal (ecoregion 4, 5, 7) to a typical Arctic community (2, 1, 8, 6, 3; Fig. 9). Investigating the composition of the most frequently occurring taxa in OBIS revealed that the taxon Diastylis rathkei was recorded in all ecoregions, followed by Campylaspis rubicunda Liljeborg, 1855, Diastylis goodsiri Bell, 1855 and Brachydiastylis resima Krøyer, 1846, which were recorded in seven of eight regions. Compared to other ecoregions of the same size, the Arctic Ocean and the area around Iceland are underrepresented in cumacean occurrence records. Currently available records in these ecoregions are restricted to 90 entries of 32 species. Within these records, the most frequently occurring species are Diastylis polaris, Sars, 1871, Leptostylis longimana, Platytyphlops semiornatus Fage, 1929, Campylaspis globosa Hansen, 1920, C. valleculate Jones, 1974, Leucon (Epileucon) spiniventris Hansen, 1920 and Platycuma holti Calman, 1905.

Figure 9 Relative family taxa composition, specimen count and predominating water masses at investigated stations of the ICECU dataset.

Each station was assigned to the predominating water mass (APW, Arctic Polar Water; ISOW, Iceland Sea Overflow Water; LSW, Labrador Sea Water; NAW, North Atlantic Water; NSAIW, Norwegian Sea Arctic Intermediate Water; NSDWc/w, cold/warm Norwegian Sea Deep Water). Circle size represents number of specimens and family taxa are distinguished by colors.

Faunistic mix in ICECU material

The morphologically and genetically investigated material of the ICECU dataset corresponded well with the earlier observed trend in the OBIS dataset of high species diversity in the North Atlantic (ecoregion 4) with 45 representative, ecoregion-specific species (Fig. 10; Table 5). With the northern extension of the NAC, the representative species number decreases to seven in ecoregion 1 (Arctic Basin) and five in ecoregion 2 (East Greenland Sea). The number of shared species occurring in more than one ecoregion decreases with distance: While the adjacent ecoregion 4 and 5 share five species, ecoregion 4 and 1 only share one species (Campylaspis intermedia Hansen, 1920). The two species Platysympus typicus G. O. Sars, 1870 and Leucon (Alytoleucon) pallidus G. O. Sars, 1865 were recorded in all ecoregions. Comparing regional distribution patterns of families, a seamless shift in the relative composition along the GIS-Ridge could be observed (Fig. 9). The highest number of investigated specimens and species was recorded in stations located south of the Ridge in the Icelandic and Irminger Basins, which have warmer and more saline water masses (ISOW, Labrador Sea Water (LSW), North Atlantic Water (NAW); ecoregion 4) and were characterized by the families Lampropidae and Nannastacidae. Stations north of the Iceland-Faroe Ridge, the Denmark Strait and on the Yermak Plateau north of Svalbard are influenced by colder and less saline water masses (APW, NSAIW, cold & warm Norwegian Sea Deep Water NSDWc & NSDWw; ecoregion 1, 2) and were mostly characterized by the families Diastylidae and Leuconidae. Representatives of the family Bodotriidae were only recorded in southern stations (Station 983, ISOW; 1,057, LSW; 1,072, NAW), as well as the Ceratocumatidae (1,057, LSW; 1,072, NAW) and the only representative specimen of the Pseudocumatidae G. O. Sars, 1878 (1,057, LSW).

Figure 10 VENN-Diagram showing the total number of representative species per ecoregion as well as shared representatives occurring within several ecoregions.

Ecoregions are sorted by the extension of the North Atlantic Current from south to north. Created using the web-tool http://bioinformatics.psb.ugent.be/webtools/Venn/.

Table 5 Table related to Fig.10: total number of representative species per ecoregion as well as shared representatives occurring within several ecoregions.

Ecoregions	Total	Representative species	
1 Arctic Basin
2 East Greenland Sea
4 North Atlantic Ocean
5 Norwegian Sea	2	Platysympus typicus, Leucon (Alytoleucon) pallidus	
1 Arctic Basin
2 East Greenland Sea	1	Diastylis spinulosa	
1 Arctic Basin
4 North Atlantic Ocean	1	Campylaspis intermedia	
2 East Greenland Sea
4 North Atlantic Ocean	4	Hemilamprops sp.1 (juv.), Hemilamprops cf. cristatus, Leucon (Macrauloleucon) spinulosus, Leptostylis ampullacea	
2 East Greenland Sea
5 Norwegian Sea	4	Hemilamprops uniplicatus, Diastylis polaris, Campylaspis sulcata, Campylaspis undata	
4 North Atlantic Ocean
5 Norwegian Sea	6	Leptostylis longimana, Diastyloides serratus, Leucon (Leucon) nathorsti, Eudorella hirsuta, Campylaspis horrida, Eudorella truncatula	
1 Arctic Basin	7	Diastylis goodsiri, Petalosarsia declivis, Leucon (Leucon) nasicoides, Diastylis rathkei, Leptostylis cf. longimana, Campylaspis rubicunda, Leucon (Leucon) aff. nathorsti	
2 East Greenland Sea	5	Leucon (Leucon) profundus, Campylaspis sp.1, Hemilamprops assimilis, Diastylis echinata, Leptostylis borealis	
4 North Atlantic Ocean	42	Bathycuma brevirostre, Campylaspides sp.1, Diastylis lucifera, Chalarostylis sp.1, Cumellopsis cf. puritani, Eudorella emarginata, Hemilamprops cf. diversus, Hemilamprops pterini, Leucon (Crymoleucon) tener, Eudorella sp.1, Diastyloides atlanticus, Hemilamprops roseus, Bodotriidae sp.1, Leptostylis sp.2, Cimmerius reticulatus, Campylaspis sp.2, Styloptocuma sp.1, Leptostylis sp.1, Leucon sp.1, Cyclaspis longicaudata B, Bodotriidae sp.2, Procampylaspis ommidion, Styloptocuma gracillimum, Leucon (Leucon) acutirostris, Leucon (Leucon) cf. robustus, Leucon (Macrauloleucon) siphonatus, Procampylaspis sp.1, Makrokylindrus (Makrokylindrus) spiniventris, Chalarostylis elegans, Styloptocuma erectum, Hemilamprops sp.2, Platytyphlops semiornatus, Nannastacidae sp.1, Campylaspis alba, Bathycuma sp.1, Styloptocuma sp. 2, Diastylis laevis, Diastyloides sp.1, Campylaspis globosa, Campylaspis costata, Pseudocuma sp.1, Cumella (Cumella) cf. decipiens	
5 Norwegian Sea	6	Iphinoe serrata, Diastyloides biplicatus, Cyclaspis longicaudata A, Mesolamprops denticulatus, Diastylis cornuta, Diastylis tumida	

In correspondence with the OBIS data, Leptostylis cf. longimana (Dia16-B; Fig. 11A) and Leptostylis ampullacea (Dia14; Fig. 11B) were the most frequently recorded species, occurring at 13 out of 21 stations, followed by Diastylis polaris (Dia06; Fig. 11C) and Leucon (Alytoleucon) pallidus (Leu05; Fig. 11D) recorded at five stations each. Other wide-ranging species, such as Platysympus typicus (Lam13; Fig. 11E) and Hemilamprops cristatus (Lam05; Fig. 11F) were found across multiple ecoregions. A majority of the species were only present in samples from one or two stations. For characteristic boreal and Arctic taxa investigated in this study, such as Leptostylis borealis (Dia15; Fig. 12A), L. (A.) pallidus (Leu05) or Hemilamprops pterini Shalla & Bishop, 2007 (Lam07; Fig. 12B), additional occurrence records were contributed within the expected ranges. The species Cimmerius reticulatus Jones, 1973 (Cer01; Fig. 12C) was the first record for the family Ceratocumatidae to be found in the North Atlantic. Interesting new records of species in Icelandic waters such as Hemilamprops cf. diversus Hale, 1946 (Lam06; Fig. 12D), previously only known from off South-eastern Australia, and Cumellopsis cf. puritani Calman, 1906 (Nan13), mostly recorded in the Mediterranean Sea, might either represent unexpected wide distribution ranges for these species or presence of morphologically closely related species, but new to science.

Figure 11 Distribution maps of species integrated in morphological and molecular analyses representing the families Diastylidae (A–C), Leuconidae (D) and Lampropidae (E–F).

Occurrences of morphologically identified species integrated in genetic analyses. Occurrence records are shown from the MAREANO and OBIS platform as well as literature data (blue), specimens morphologically investigated in this study (orange) and subsequently genetically investigated (grey triangle with sequence ID) and type and/or syntype locality of putative species (yellow star with reference literature). Genetic lineages are highlighted with dotted circles and separated by assigned letters A–C.

Figure 12 Distribution maps of species integrated in morphological and molecular analyses representing the families Diastylidae (A, E), Lampropidae (B, D), Ceratocumatidae (C) and Nannastacidae (F).

In order to minimize the taxonomic knowledge gap, especially in ecoregion 2, the focus of this study is on specimens collected within the regions 1, 2, 4 and 5, which represent all three water-mass categories (North Atlantic, Subarctic, Arctic) and differed remarkably in their taxa composition.

Representatives of the Arctic Ocean (ecoregion 1)

One representative specimen of the circumpolar species Diastylis spinulosa Heller, 1875 (Dia08; Fig. 12E) was detected from north of Spitzbergen. In congruence with the OBIS occurrence records, ecoregion 1 was dominated by representatives of the families Diastylidae, Leuconidae and Nannastacidae. The morphologically examined material from the Arctic Ocean included only four species: Leptostylis cf. longimana (Dia16-B) dominated in specimen number, followed by Campylaspis rubicunda (Nan08; Fig. 12F), Campylaspis intermedia (Nan06; Fig. 13A) and one specimen of Leucon (Alytoleuco) pallidus (Leu05). Campylaspis intermedia is known to be widely distributed in the whole Atlantic Ocean but was not recorded in the Arctic yet. Thus, the occurrence records in the present study from north of Svalbard extend the previously assumed distribution range significantly. Campylaspis rubicunda is known from the North Pacific, the North Atlantic and the Arctic Oceans and Leptostylis longimana is widely distributed in the cold areas of the Northern Atlantic. Nevertheless, none of the previously mentioned species is found strictly in ecoregion 1, but rather widely distributed and occurring in other ecoregions.

Figure 13 Distribution maps of species integrated in morphological and molecular analyses representing the families Nannastacidae (A, C), Leuconidae (B), Diastylidae (D) and Bodotriidae (E).

Representatives of East Greenland Sea (ecoregion 2)

The most common species in ecoregion 2 was Diastylis polaris (Dia06), which was sampled at five IceAGE stations composed of NSDWw and NSDWc. This observation corresponds with the occurrence records found in OBIS, which also include the morphologically closely related Diastylis stygia G. O. Sars, 1871, both being described as true Arctic species. The species Leptostylis borealis (Dia15), originally described south of Franz-Josef-Land, was recorded for the first-time off Iceland. Most representatives of the family Leuconidae in the IceAGE material were sampled in ecoregion 2, north of the Iceland-Faroe Ridge. This result suggested a restricted distribution range of sampled representatives of the family Leuconidae to colder and fresher water masses (e.g., APW, NSAIW and NSDW). The species Leucon (Leucon) profundus Hansen, 1920 (Leu11; Fig. 13B) and Leucon (Alytoleucon) pallidus (Leu05) were the most common species of this family in the investigated material. Vassilenko (1989) and Gerken & Watling (1999) described both with a circumpolar distribution range from the North Atlantic Ocean to the Canadian Arctic Ocean, but mostly found in cold-water stations (Watling & Gerken, 2005). However, by our samples, new occurrence records of these species were added in the Norwegian Sea waters influenced by the North-Atlantic current, suggesting an extension of the distribution range for warmer waters of the previously assumed cold-water species.

Representatives of the North Atlantic and Norwegian Sea (ecoregion 4 and 5)

Ecoregion 4 was dominated in specimens by Hemilamprops cf. diversus (Lam07) and Hemilamprops pterini (Lam08), which are taxa found predominantly in warmer waters, as well as Campylaspis sp. 2 (Nan10; Fig. 13C). In general, the genera Hemilamprops and Campylaspis reached highest record numbers both in terms of taxa and specimen count. The widely distributed boreal Atlantic species Hemilamprops cf. cristatus (Lam05-B) is recorded to occur in high abundances within ecoregion 4 and 5, though it was also sampled at one station in the Denmark Strait. Specimens in ecoregion 4 were mostly sampled in deep-sea habitats, while specimens in ecoregion 5 were mostly sampled in coastal and fjord habitats. Typical Atlantic Ocean species of other families only recorded south of the GIS-Ridge in warmer waters were Diastyloides atlanticus Reyss, 1974 (Dia10; Diastylidae; Fig. 13D), Bathycuma brevirostre Norman, 1879 (Bod01; Bodotriidae) and Cyclaspis longicaudata Sars, 1865 (Bod05; Bodotriidae; Fig. 13E), as well as Pseudocuma sp. 1 (Pse02), the only representative of the Pseudocumatidae in the ICECU dataset. Cimmerius reticulatus (Cer01), expanding its distribution northward from the Bay of Biscay, is the only representative of the Ceratocumatidae. The specimens of Eudorella truncatula sampled in ecoregion 4 (Leu04-A/-B; Fig. S2-U) and ecoregion 5 (Leu04-C) were all separated into distinct lineages, which was also observed in Cyclapsis longicaudata from ecoregion 4 (Bod05-B) and ecoregion 5 (Bod05-A).

Discussion

Combined species delimitation approach - morphology and genetics

As only a handful of studies applied molecular methods for species delimitation within cumaceans, the combined approach in this study highlights the high quality and overall congruence in most cases of morphological and genetic analyses. Even though delimiting species based solely on one mitochondrial marker like 16S rDNA has been questioned as there is no universal threshold for species delimitation (Meyer & Paulay, 2005; Meier et al., 2006; Wiemers & Fiedler, 2007; Schwentner, Timms & Richter, 2011; Collins & Cruickshank, 2013), in our study a clear barcoding gap between 2% and 8% was observed among most inferred species, corresponding nicely with morphological taxonomic characters in the vast majority of cases. The 4% genetic distance observed between the geographically widely separated (>3,000 km) Diastylis rathkei (Dia07) individuals cannot be easily interpreted as either intra- or interspecific and might hint at a recent and/or ongoing speciation event. Similar observations have been made previously for 16S rDNA of cumaceans and other peracarids like isopods (Brökeland & Raupach, 2008; Held & Wägele, 2005; Rehm, 2007; Rehm et al., 2020; Riehl, Lins & Brandt, 2018). Herein, intraspecific distances were usually below 1%, though geographically widely distributed species featured up to 3% or 5%, respectively. Conversely, interspecific distances exceeded 7%. Also, here the higher intraspecific genetic distances of 5% were tentatively suggested to represent potential cryptic species (Rehm, 2007). Based on these findings we suggest that the rare cases of conflict between morphological and genetic data represent cases of cryptic diversity or extensive morphological variability.

Examples of morphologically cryptic diversity

The incongruence of 16S sequence data revealing high genetic diversity within a species (or even cryptic species) was observed in six cases (Table 4). This affects Hemilamprops cristatus (Lam05, A–B), Eudorella truncatula (Leu04, A–C), Cyclaspis longicaudata (Bod05, A–B), Leptostylis borealis (Dia15, A–B) and Leptostylis sp. 1 (Dia18, A–B; Fig. S2L). The intraspecific genetic distances observed within each of these taxa greatly exceeded those commonly observed within species of Cumacea and other peracarids. We therefore conclude that all of these cases indicate presence of morphologically cryptic species. Moreover, some of these cryptic species were not even recovered as sister species, but widely separated (Fig. 2). Further taxonomic studies will be needed to prove these cases.

In Cyclaspis longicaudata (Bod05, A–B), Eudorella truncatula (Leu04, A–C) and Leptostylis borealis (Dia15, A–B) the respective cryptic species were geographically well separated and usually occurred in different water masses. In the case of Hemilamprops cristatus (Lam05, A–B), morphological re-examinations showed weak differences in the rostrum length and the serration of the dorsal crest on the carapace (shorter rostrum and smaller, but more teeth in Lam05-A). However, one specimen (ICE1-Lam018) sampled on the Greenland slope in Subarctic waters and morphologically corresponding to Lam05-B, clustered genetically together with Lam05-A from the Norwegian continental shelf.

Our cumacean examples do support the finding in other peracarid taxa (amphipods and isopods) of either overestimations (Lörz et al., 2020) or underestimations of intraspecific divergence (Brix, Svavarsson & Leese, 2014b; Jennings, Golovan & Brix, 2019; Paulus et al., in press). This emphasizes that sampling from a geographically limited portion of a species’ range only risks missing relevant genetic variation, which blurs an important line between species-level and population-level diversity. Molecular species delimitation should, thus, include specimens sampled in the widest possible distribution range of the examined taxon (Knox, 2012).

Examples of morphological variability and taxonomic incongruence

Specimens identified as Diastylis polaris (Dia06) and Diastylis stygia were identical in 16S. Interestingly, Zimmer (1926) synonymized these species based on Ohlin’s (1901) and Stebbing’s (1913) conclusion of their conspecifity. In Zimmer, 1980 re-examined a specimen of D. stygia collected by the Russian Sadko Expedition (Zimmer, 1943) and separated it again from D. polaris as a valid species. Our study lays additional support for D. stygia being a synonym of D. polaris. Similarly, after re-examination of Platysympus typicus (Lam14) sampled off East Greenland and P. tricarinatus Hansen, 1920 from the Norwegian shelf, these two species names are probably representing the same species. Gerken (2018) called the assumed differentiating characters of more or less conspicuous folds on the carapace into doubt and suggested that less prominent characters might be owed to juvenile stages. Therefore, P. tricarinatus is assumed to be a synonym of P. typicus. Similarly, Leucon (Leucon) aff. nathorsti (Leu09) differed morphologically from L. (L.) nathorsti (Leu08) by the presence of two dorsolateral teeth on the frontal lobe of the carapace and a more acute rostrum and, thus, identified as a possible separate species. However, in this 16S analyses these two morphotypes proved to be identical, suggesting the presence of a single, morphologically variable species. As Leu08-specimens were sampled on the Norwegian continental shelf in North Atlantic waters and Leu09 was collected off Svalbard in Subarctic waters, an ecologically-driven morphological population variation might be implied.

Further, we found that the two morphologically almost indistinguishable species Leptostylis longimana (Dia16-B) and L. ampullacea (Dia14) grouped into a large unresolved clade forming a “Leptostylis longimana/ampullacea” species complex. Species in the genus Leptostylis are rather difficult to distinguish as there is a certain degree of phenotypic plasticity tied to sex and growth stages, and some morphological distinctions are quite subjective, such as ‘clumsier’ body form of L. ampullacea compared to L. longimana (Sars, 1900). A second “Leptostylis longimana”-clade was retrieved, based on specimens from coastal Norway, genetically well separated from the Iceland/Arctic “L. longimana/ampullacea”-clade by 26–27% p-distance. Based on the present data, the Iceland/Arctic specimens should be referred to L. ampullacea, originally described from Kullaberg (off Sweden). This further implies that the “true” L. longimana, originally described from the Oslofjord, a more coastal and/or southern distribution. Further studies including additional specimens and gene markers, as well as museum type material will be necessary to resolve the taxonomy in more detail.

Biogeographic integration

Data-mining implications

This study contributed with the ICECU dataset first occurrence records of 18 species representing five families within the investigated ecoregions (Fig. 8). Additionally, about 25% of morphologically determined taxa could not be assigned to species level, which might either constitute known species from originally other regions or new species to science. The extension of distribution ranges of ecoregional representative species clearly shows our knowledge gaps on estimated distribution patterns of cumaceans. By monitoring the impact of a changing climate on species distributions based on time series of the first occurrences of this species, the benefits of publicly accessible distribution data of marine animals on platforms such as OBIS are undeniable. The linkage to WoRMS ensures verified taxonomic name information following the Darwin Core standard and connection to other sources (Costello et al., 2007; Wieczorek et al., 2012). Still, the determination of species demands knowledge of taxonomy, ecology, and morphological characters of the investigated taxon. Especially molecular species delimitation depends on prior morphological identification and database confidence. The importance of such reliable species name assignments was especially observed in the genus Leptostylis, in which hidden diversity was found when integrating genetic data. Thus, caution should be taken when using public distribution databases due to their restricted possibilities to present hidden diversity. This study showed that species identifications of a large dataset based only on morphological delimitation may underestimate true diversity. For example, in the case of the species Hemilamprops cristatus and Eudorella truncatula, which are assumed to be a widely distributed species in the boreal Arctic as well as in the North American basin (Watling, 2009), genetic analyses revealed either a cryptic speciation due to geographical separation and different water-mass conditions or species from distinct populations with separated geographical origins.

Are ecoregions reflected in species distribution?

The results of this study support the suggestion of Hansen (1920) and Watling & Gerken (2005) that water-mass characteristics are an important controlling variable for cumacean species occurrences. The community composition was observed to change from warm-water dominating families in ecoregion 4 south of the GIS-Ridge (Lampropidae, Bodotriidae, Nannastacidae) to families dominating in colder and less saline Subarctic and Arctic water masses found in ecoregion 1 and 2 (Diastylidae, Leuconidae; Fig. 9).

Closer investigation of cumacean distributions on species level revealed that most species occurred in multiple ecologically similar ecoregions (Fig. 10; Table 5). Even though typical ecoregion-specific representatives could be determined, in many cases these also occurred in other ecoregions. This pattern was also corroborated by the genetic data. For example, Vassilenko (2002) categorized the species Campylaspis globosa (Nan04) and Hemilamprops uniplicatus G. O. Sars, 1872 (Lam11) as widely distributed Arcto-Atlantic bathyal species of Atlantic origin. This study supported the preceding assumption, as specimens inhabiting Arctic and Subarctic waters in the Denmark Strait (ecoregion 2) and Atlantic waters on the Norwegian continental shelf (ecoregion 4 & 5) were morphologically and genetically identical within the species (Figs. S2Q, S2α). The same case was observed in Platysympus typicus (Lam13) and Diastylis polaris (Dia06) from Atlantic to Arctic water influenced ecoregions (1, 2, 4, 5; Figs. 11C, 11E).

Some species revealed hidden diversity reflected by patchy distribution patterns within ecoregions. Cyclaspis longicaudata Bod05-B was sampled in the Iceland Basin (ecoregion 4) and was genetically differentiated and geographically separated by the GIS-Ridge to Bod05-A from the Norwegian continental shelf (ecoregion 5; Fig. 13E). A similar case was observed between the species Hemilamprops cristatus Lam05-B from a deep Iceland Basin station (2,500 m) and Lam05-A, which was sampled close to the type locality from Skagerrak (700 m depth; Fig. 11F). As this species is reported as a widely distributed boreal Atlantic species, cryptic speciation might be considered due to the geographical separation by the GIS-Ridge. It would seem that H. cristatus in the Iceland basin might be a hidden, putative species new to science. Simultaneously, the Leptostylis longimana/ampullacea complex Dia14/-16-B from cold-water masses (NSDWw, APW, APW/NSAIW) was revealed to be widely distributed from Iceland to Arctic regions (ecoregions 1 and 2) over a distance of more than 2,500 km. Dia16-A, though, was revealed to be a distinct genetically differentiated population in ecoregion 4. Earlier records of this species complex are from the same distribution area, and described as a predominantly bathyal-Atlantic, boreal-Arctic species (Jones, 1976; Vassilenko, 1989). Watling & Gerken (2005) observed L. longimana to be present over a wide temperature range. Another example in ecoregion 2 for potentially disrupted gene flow between populations by geographical barriers is the species Leptostylis borealis Dia15-A from the East Greenland shelf and Dia15-B from East Iceland Norwegian Sea, separated by the Denmark Strait (Fig. 12A). In contrast, Leucon (A.) pallidus (Leu05) sampled at these stations showed no genetic differentiation between specimens, despite the Denmark Strait as a potential barrier. For the investigation of distinct cumacean distribution patterns as proposed by Watling & Gerken (2005), a larger sample size is needed as many morpho- and genospecies were represented as singletons or were sampled in higher numbers, but at solely one station.

Conclusions

This study confirmed the advantage of a combined approach of traditional morphological and modern molecular techniques to delimit cumacean species and uncover hidden diversity, compared to delimitations based solely on one method. Some species may need more taxonomic attention and re-evaluation. We have shown examples of underestimated diversity as well as overestimated diversity. However, the advantage of molecular investigations for testing important questions of species diversity correlates significantly with prior species identifications and emphasizes the importance of a robust basis of taxonomic knowledge and morphological examination.

For example, in ecoregion 2, 98% of the specimens could only be determined to order level. When the resolution of identification only gets to order level, the species occurrence data is influenced as the species level information is not shown in public databases (for example). Thus, with more species level identifications in ecoregion 2, it is a high likelihood to find more species new to science or even more ecoregion-representative species. While this ecoregion was characterized by five representative species, ecoregion 4 revealed high species diversity with 45 representative species correlating with the highest sampling effort of all investigated regions.

As for other peracarid groups, the GIS-Ridge plays an important role as a geographical barrier and separates ecoregion-specific cumacean communities between the North Atlantic Ocean in the south and the Subarctic seas in the north. Although the biogeographic results of this study furthermore support the assumption of earlier studies that water-mass characteristics are important controlling variables for cumacean species occurrences, this remains a hypothesis unless a more detailed ecological observation of factors shaping cumacean distribution with statistical analyses including not only water masses, but further abiotic factors (e.g., depth, sedimentary characteristics, potential geographical barriers) is undertaken.

Supplemental Information

Supplemental Information 1 ABGD frequency spectrum of p-distances.

Bar plot shows a clear barcoding gap based on the applied threshold of P = 0.01–0.1 between intra- and interspecific variation for all families: (A) Leuconidae (intra: 0–0.01; inter: 0.17–0.38); (B) Bodotriidae and Nannastacidae (0–0.01; 0.08–0.45); (C) Diastylidae and Pseudocumatidae (0–0.04; 0.15–0.37); (D) Ceratocumatidae and Lampropidae (0-0.02; 0.13-0.34).

Click here for additional data file.

Supplemental Information 2 Distribution maps of species integrated in morphological and molecular analyses representing the families Bodotriidae (A–C) and Diastylidae (D–F).

Occurrence records are shown from the MAREANO and OBIS platform as well as literature data (blue), specimens morphologically investigated in this study (orange) and subsequently genetically investigated (grey triangle with sequence ID) and type and/or syntype locality of putative species (yellow star with reference literature).

Click here for additional data file.

Supplemental Information 3 Distribution maps of species integrated in morphological and molecular analyses representing the family Diastylidae (G–L).

Occurrence records are shown from the MAREANO and OBIS platform as well as literature data (blue), specimens morphologically and subsequently genetically investigated (grey triangle with sequence ID) and type and/or syntype locality of putative species (yellow star with reference literature). Genetic lineages are highlighted with dotted circles and separated by assigned letters A–B.

Click here for additional data file.

Supplemental Information 4 Distribution maps of species integrated in morphological and molecular analyses representing the family Lampropidae (M–R).

Occurrence records are shown from the MAREANO and OBIS platform as well as literature data (blue), specimens morphologically investigated in this study (orange) and subsequently genetically investigated (grey triangle with sequence ID) and type and/or syntype locality of putative species (yellow star with reference literature).

Click here for additional data file.

Supplemental Information 5 Distribution maps of species integrated in morphological and molecular analyses representing the family Leuconidae (S–X).

Occurrence records are shown from the MAREANO and OBIS platform as well as literature data (blue), specimens morphologically and subsequently genetically investigated (grey triangle with sequence ID) and type and/or syntype locality of putative species (yellow star with reference literature). Genetic lineages are highlighted with dotted circles and separated by assigned letters A-C.

Click here for additional data file.

Supplemental Information 6 Distribution maps of species integrated in morphological and molecular analyses representing the families Leuconidae (Y) and Nannastacidae (Z–δ).

Occurrence records are shown from the MAREANO and OBIS platform as well as literature data (blue), specimens morphologically investigated in this study (orange) and subsequently genetically investigated (grey triangle with sequence ID) and type and/or syntype locality of putative species (yellow star with reference literature).

Click here for additional data file.

Supplemental Information 7 Distribution maps of species integrated in morphological and molecular analyses representing the families Nannastacidae (ε) and Pseudocumatidae (ζ).

Occurrence records are shown from the MAREANO and OBIS platform as well as literature data (blue), specimens morphologically investigated in this study (orange) and subsequently genetically investigated (grey triangle with sequence ID) and type and/or syntype locality of putative species (yellow star with reference literature).

Click here for additional data file.

Supplemental Information 8 Definition of water masses.

Salinity and temperature ranges for water mass identifications according to (1) Schlichtholz & Houssais (2002) and (2) Hansen & Østerhus (2000) mentioned in this study and used as baseline definitions for a T-S-plot of PASCAL and IceAGE expedition CTD-data.

Click here for additional data file.

Supplemental Information 9 Taxonomy of morphologically investigated taxa of the order Cumacea with their assigned species ID.

Click here for additional data file.

Supplemental Information 10 Data source and station information on specimens incorporated in the distribution maps.

Click here for additional data file.

Supplemental Information 11 Complete list on all morphologically and genetically investigated specimens in this study.

Morphological identification of 947 investigated specimens, resulting in 77 putative species. Field ID was assigned to each specimen during DNA extraction. Out of 123 extracted specimens, 80 yielded sequence data of sufficient quality to be included in the molecular species delimitation (highlighted in bold).

Click here for additional data file.

Supplemental Information 12 Dataset 2-ABGD groups of the Leuconidae.

Uncorrected intra- and interspecific pairwise genetic distance range (p-distance) of putative species of the Leuconidae, delimited ABGD groups based on the applied threshold of P = 0.01–0.1 (12 groups) and the groups’ nearest neighbor.

Click here for additional data file.

Supplemental Information 13 Dataset 2-Genetic distances (uncorrected p-distances) among 16S rRNA putative species of the family Leuconidae calculated in MEGA X.

Click here for additional data file.

Supplemental Information 14 Dataset 3-ABGD groups of the Bodotriidae and Nannastacidae.

Uncorrected intra- and interspecific pairwise genetic distance range (p-distance) of putative species of the cumacean families Bodotriidae and Nannastacidae, delimited ABGD groups based on the applied threshold of P = 0.01–0.08 (13 groups) and the groups’ nearest neighbor.

Click here for additional data file.

Supplemental Information 15 Dataset 3-Genetic distances (uncorrected p-distances) among 16S rRNA putative species of the families Bodotriidae and Nannastacidae calculated in MEGA X.

Click here for additional data file.

Supplemental Information 16 Dataset 4-ABGD groups of the Diastylidae and Pseudocumatidae.

Uncorrected intra- and interspecific pairwise genetic distance range (p-distance) of putative species of the cumacean families Diastylidae and Pseudocumatidae, delimited ABGD groups based on the applied threshold of P = 0.01–0.1 (17 groups) and the groups’ nearest neighbor.

Click here for additional data file.

Supplemental Information 17 Dataset 4-Genetic distances (uncorrected p-distances) among 16S rRNA putative species of the families Diastylidae and Pseudocumatidae calculated in MEGA X.

Click here for additional data file.

Supplemental Information 18 Dataset 5-ABGD groups of the Ceratocumatidae and Lampropidae.

Uncorrected intra- and interspecific pairwise genetic distance range (p-distance) of putative species of the cumacean families Ceratocumatidae and Lampropidae, delimited ABGD groups based on the applied threshold of P = 0.01–0.1 (13 groups) and the groups’ nearest neighbor.

Click here for additional data file.

Supplemental Information 19 Dataset 5-Genetic distances (uncorrected p-distances) among 16S rRNA putative species of the families Ceratocumatidae and Lampropidae calculated in MEGA X.

Click here for additional data file.

Supplemental Information 20 Sequence alignments used for genetic analyses.

Click here for additional data file.

Firstly, we would like to emphasize that the base of this study is formed by excellent international collaboration, fruitful discussions, and trust-based exchange of experience and knowledge. It would not have been possible without team- and networking, starting with those, who invest many days to sample at sea, preserve, catalogue and sort benthic samples for further analyses. We owe an immense debt to Karen Jeskulke, Nicole Gatzemeier, Antje Fischer, Sven Hoffmann and the whole team of DZMB HH for the technical support at all levels. We warmly thank Kathrin Philipps-Bussau, Petra Wagner, Nancy Mercado Salas and Alexandra Kerbl for maintenance of the CeNak collection material. Many thanks to Simon Bober, Johanna Bober, Mariam Dunker and Oliver Hawlitschek for the help on laboratory issues. Our sincere thanks go to the Captains and Crew of RV Meteor and RV Polarstern during M85/3 (IceAGE) and PS106/1 (PASCAL). Big thanks to Team ‘SIEMO’ for their help and efficient work at sea. Special thanks go to Sarah Gerken and Ute Mühlenhardt-Siegel for taxonomical expertise and advice. We are grateful to Louise Lindblom for guidance and assistance in the Biodiversity laboratory at University of Bergen.

Additional Information and Declarations

Competing Interests

Author Contributions

Field Study Permissions

DNA Deposition

Data Availability

The authors declare that they have no competing interests.

Carolin Uhlir conceived and designed the experiments, performed the experiments, analyzed the data, prepared figures and/or tables, authored or reviewed drafts of the paper, and approved the final draft.

Martin Schwentner conceived and designed the experiments, analyzed the data, authored or reviewed drafts of the paper, and approved the final draft.

Kenneth Meland conceived and designed the experiments, analyzed the data, authored or reviewed drafts of the paper, and approved the final draft.

Jon Anders Kongsrud conceived and designed the experiments, authored or reviewed drafts of the paper, and approved the final draft.

Henrik Glenner conceived and designed the experiments, authored or reviewed drafts of the paper, and approved the final draft.

Angelika Brandt conceived and designed the experiments, authored or reviewed drafts of the paper, and approved the final draft.

Ralf Thiel conceived and designed the experiments, authored or reviewed drafts of the paper, and approved the final draft.

Jörundur Svavarsson conceived and designed the experiments, authored or reviewed drafts of the paper, and approved the final draft.

Anne-Nina Lörz conceived and designed the experiments, authored or reviewed drafts of the paper, and approved the final draft.

Saskia Brix conceived and designed the experiments, analyzed the data, authored or reviewed drafts of the paper, and approved the final draft.

The following information was supplied relating to field study approvals (i.e., approving body and any reference numbers):

Field permits are available with the following cruise codes: M85/3 (IceAGE; BR3843/3-1) and PS106/1 (PASCAL; AWI_PS106_00).

The following information was supplied regarding the deposition of DNA sequences:

Aligned sequences are available in the Supplemental File. The ICECU BoLD project is available at: dx.doi.org/10.5883/DS-ICECU.

The sequences included in this study are available at GenBank: HQ450558, MZ402659, MZ402660, AJ388111, MZ402681, MZ402680, MK613872.1, MK613873.1, HQ450557, MK613886.1, MZ402679, MZ402678, MZ402677, MK613898.1, MK613897.1, MK613904.1, MK613901.1, MK613911.1, MZ402685, MZ402686, MZ402683, MZ402682, MZ402684, MZ402687, MK613902.1, MK613903.1, MK613905.1, HQ450555, U81512, MK613906.1, MK613899.1, MK613900.1, MZ402689, MZ402688, MK613910.1, MK613907.1, MK613909.1, MK613908.1, HQ450556, MZ402704, MZ402711, MZ402708, MZ402710, MZ402709, MZ402707, MZ402706, MZ402705, MZ402712, MZ402713, MK613921.1, MK613922.1, MZ402723, MZ402714, MZ402722, MZ402717, MZ402718, MZ402716, MZ402719, MZ402720, MZ402721, MZ402724, MZ402715, MZ402702, MZ402701, MK613925.1, MZ402676, MZ402675, MK613924.1, MK613913.1, MK613914.1, MZ402695, MZ402696, MZ402697, MZ402692, MZ40269, MZ402694, MZ402691, MZ402698, MZ402699, MK613923.1, MZ402690, MZ402700, MK613915.1, MK613916.1, MK613917.1, MZ402737, MZ402736, MK613918.1, MK613919.1, MK613870.1, MK613887.1, MK613888.1, U81513, MK613881.1, MK613882.1, MK613884.1, MK613883.1, MK613885.1, MZ402728, MZ402729, MZ402731, MZ402730, MZ402725, MZ402726, MZ402727, MK613892.1, MK613891.1, HQ450522, HQ450543, HQ450549, HQ450550, HQ450537, MK613889.1, HQ450551, HQ450552, HQ450553, MK613895.1, MK613893.1, MK613894.1, MK613890.1, MZ402734, MZ402732, MZ402733, MZ402735, HQ450554, MK613876.1, MK613874.1, MZ402662, MZ402663, MK613877.1, MZ402664, MZ402666, MZ402665, MZ402670, MZ402669, MZ402667, MZ402668, MZ402671, MZ402661, MZ402672, MZ402674, MZ402673, MK613875.1, MK613878.1, MZ402738, MK613871.1, AJ388110, DQ305106, AF260869, AF260870, AY693421, AF259533, DQ305111, MK813124.

The following information was supplied regarding data availability:

Occurrence data of the ICECU dataset is available at the OBIS platform: Uhlir, C.; Meland, K.; Glenner, H.; Brandt, A.; Brix, S. (2021). North Atlantic and Arctic Cumacea sampled during the IceAGE (2011) and PASCAL (2017) project. Marine Data Archive.

http://ipt.vliz.be/eurobis/resource?r=cumacea_pascal_iceage.

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
