# Peer review of "Adding pieces to the puzzle: insights into diversity and distribution patterns of Cumacea (Crustacea: Peracarida) from the deep North Atlantic to the Arctic Ocean"

_PeerJ, doi:10.7717/peerj.12379_

## Round 0.1 · original submission · Minor Revisions

This paper reflects a thorough study of cumacean crustaceans in the Nordic Seas, using both morphological and molecular approaches. The authors have done a great job in discussing the congruences and incongruences between the two approaches. While using only one genetic marker incurs limitations for interpretation of the results, the dataset analyzed is vast and the analyses are comprehensive.

Both reviewers recognized the high quality and importance of this paper and had mostly minor (style) comments. Nevertheless, they had some more substantial issues to be addressed. Review 1 pointed out that the reason that almost all families were not recovered as monophyletic is likely due to the use of 16S, which is not appropriate for deeper phylogeny. Reviewer 2 suggests the use of a tree-based species delimitation method (eg, mPTP, GMYC) in addition to the pairwise method used by the authors (with the exception of singletons). Please refer to these issues and the rest of the comments in the revision.

·

Basic reporting

1. Line 36 and 330, phrase "all seven known families", as correctly acknowledged in other places in the article, there are 8 accepted families of Cumacea. If you mean the seven known from the North Atlantic, state that, because as it is written it is incorrect.
2. Line 105, the family Nannastacidae is not particularly predominantly a warm water group, it is globally distributed including at both poles and in deep water, do you have a citation for this assumption? Bodotriidae is more of a warm water family.
3. Line 190, why "so-called Brenke sled", why not just Brenke Sled?
4. Line 339, families not recovered as monophyletic: using primarily a short fragment of highly variable 16S, that is not a surprise. One does not expect to recover deeper phylogeny with a single, highly variable, short fragment that is appropriate for assessing population/species level relationships. Please remove this statement or amend it to point out that one would not expect to recover them as monophyletic with this fragment, as it might be used to suggest that all the families are not monophyletic, which is not a reasonable conclusion based on this data.
5. Line 668, "support the assumption", it is not an assumption, it is a suggestion. Assumption means a thing that is accepted as true without proof or evidence, which is not the situation being described. Hansen's suggestion was based on observation, and our suggestion (Watling & Gerken 2005) was based on observation and analysis of species distributions.

Experimental design

no comment

Validity of the findings

This is an important contribution to cumacean systematics, and as such should be published.

Reviewer 2 ·

Basic reporting

The writing style is in general good, as is the English. All manuscripts (written by native and non-native English speakers alike!) contain grammatical mistakes, but these are to be expected. I thank the authors for what are, in the vast majority, clearly written sentences with easy to follow logic and argumentation. Specific line-by-line corrections are provided in the attached PDF. The article structure, figures, and tables are all sufficient. Data have not been shared yet, but this can be easily fixed. The results are relevant to the hypotheses.

Experimental design

This is original primary research of proper scope, with a well-defined, relevant, and meaningful question. It definitely expands our knowledge and fills a critical research gap. Analyses are in general well-conducted and sufficient, with one suggested addition (item 2 in the attached PDF)

Validity of the findings

Conclusions (meaning explaining why the observed results might have occurred, what they tell us about a larger picture, or why they are interesting/important) are often lacking. The potential is absolutely there, and the authors have already included basically all of the introductory/background information needed for the reader to glean this understanding. Specific examples with suggestions to expand/improve are given in the attached PDF.

Additional comments

Please see the attached PDF for general and specific comments. A lot of really good, thoughtful, and well-done work went into this project, which is admirably large in geographic scope and overarching "vantage point". It may seem that many pieces of the ms were critiqued, but it is definitely a very worthy work which deserves some time spent to improve it, so it can have the impact on readers it deserves. Overall a very good effort that just needs a little more attention in some regards!

Annotated reviews are not available for download in order to protect the identity of reviewers who chose to remain anonymous.

---

## Round 0.2 · accepted · Accept

The authors have done extensive work of addressing all reviewer comments and embedding necessary changes in the manuscript. The extent of the work depicted in this paper is impressive, and its importance mainly lies in the comparison of morphological and molecular species delimitation. I am looking forward to seeing it published.

ddjr